# Long-term monitoring of ultratrace nucleic acids using tetrahedral nanostructure-based NgAgo on wearable microneedles

Bin Yang[1], Haonan Wang[1], Jilie Kong [ID][1] & Xueen Fang [ID][1] ✉

Real-time and continuous monitoring of nucleic acid biomarkers with wearable devices holds potential for personal health management, especially in the context of pandemic surveillance or intensive care unit disease. However, achieving high sensitivity and long-term stability remains challenging. Here, we report a tetrahedral nanostructure-based Natronobacterium gregoryi Argonaute (NgAgo) for long-term stable monitoring of ultratrace unamplified nucleic acids (cell-free DNAs and RNAs) in vivo for sepsis on wearable device. This integrated wireless wearable consists of a flexible circuit board, a microneedle biosensor, and a stretchable epidermis patch with enrichment capability. We comprehensively investigate the recognition mechanism of nucleic acids by NgAgo/guide DNA and signal transformation within the Debye distance. In vivo experiments demonstrate the suitability for real-time monitoring of cell-free DNA and RNA with a sensitivity of 0.3 fM up to 14 days. These results provide a strategy for highly sensitive molecular recognition in vivo and for on-body detection of nucleic acid.

Recently developed genetic testing techniques allow the identification of related diseases through nucleic acid-based diagnostics[1–3]. Among these established tools, PCR[4], isothermal nucleic acid amplification coupled with CRISPR technology[5], and whole genome sequencing[6] have been optimized and possessed the ability to provide physiological information indicative of numerous pathologies over the past decades. Despite great advances, PCR and genome sequencing are confronted with limitations, such as long turnover time and bulky equipment. More recently, some methodologies have been rapidly developed based on amplification-free strategies for the detection of unamplified target nucleic acids, including multiple CRISPR-effectors[7,8], cascade CRISPR reactions[9], aptamer-linked graphene field effect transistors[10], engineered LwaCas13a[11], and strand displacement reactions[12]. Such new approaches might address limitations of traditional nucleic acid molecular diagnosis. However, in some circumstances, amplification-free approaches require hand-free, sensitive, continuous monitoring, which might be approached by fully integrated wearable techniques.

Emerging wearable technology has substantially provided transformative technology for continuous and simultaneous monitoring of a wide range of biomarkers on the human epidermis, which has expanded their utility in the fields of personalized medicine, internet-of-medical-things, and real-time disease diagnosis[13–16]. It is worth noting that state-of-the-art wearable devices have gained much attention for their ability to monitor subjects' physiological information, for example, tracking deep tissue with artificial intelligence[17], imaging cardiac functions[18], detecting chronic infectious wounds[19], evaluating body thermoregulation[20], and detecting tumor regression[21]. Some of these are already on the commercial market for glucose, body water loss, and sweat electrolyte monitoring[22]. Hence, wearables might provide an alternative approach for continuous monitoring, e.g., intensive care unit (ICU) patients with sepsis. However, the current version of wearables is hardly available for ICU scenarios because it still has some limitations to real-time and long-term monitoring of low-abundance disease-related cell-free DNA (cfDNA) or RNA biomarkers, which are approximately 10 to 15 ng per milliliter on average[23].

In this study, we report fully integrated wearable electronics based on tetrahedral DNA nanostructures (TDNs) and prokaryotic argonaute technology for universal nucleic acid real-time monitoring and sepsis-associated intervention caused by EBV, staphylococcus

[1]Department of Chemistry and Institutes of Biomedical Sciences, Fudan University, Shanghai 200433, PR China. ✉e-mail: fxech@fudan.edu.cn

aureus (SA) and pseudomonas aeruginosa (PA). The wireless wearable system consists of a flexible circuit board, an engineered microneedle, and a stretchable epidermis patch. TDN scaffolds[24-28] were used to realize ultrasensitive detection, while the engineered Natronobacterium gregoryi Argonaute (NgAgo) biorecognition interface guided by single-stranded DNA facilitated long-term stability of the system. The integrated wearable system is employed for real-time tracking of sepsis-related cfDNA and RNA in interstitial fluid (ISF) and therefore expanded possibility in monitoring sepsis for ICU onset.

## Results

### Design, assembly, and mechanism of the fully integrated wearable electronics

A wearable system with high sensitivity and stability is designed for long-term and real-time monitoring of ultratrace nucleic acids, including DNA and RNA, based on the prokaryotic argonaute technology (Natronobacterium gregoryi Argonaute, NgAgo) and TDN. Figure 1 details the wearable electronic system, which is accessible to record changes in unamplified target nucleic acids continuously and with high sensitivity over a period of time (more than two weeks). As shown in Fig. 1a, this integrated wearable system consists of a disposable microneedle patch and reusable electronics (wireless flexible circuit board, functional thermoplastic polyurethane film). In this study, the disposable microneedle (MN) patch was made of a three-electrode array in a compact manner, namely, a three-in-one patch. Then, a functional thermoplastic polyurethane (TPU) consisting of a first layer of silver and a second layer of carbon nanotubes was prepared by spray printing and casting technology. The functionalized TPU film with anode and cathode patterns was individually connected with a wireless flexible circuit board (FCB), which was able to enrich

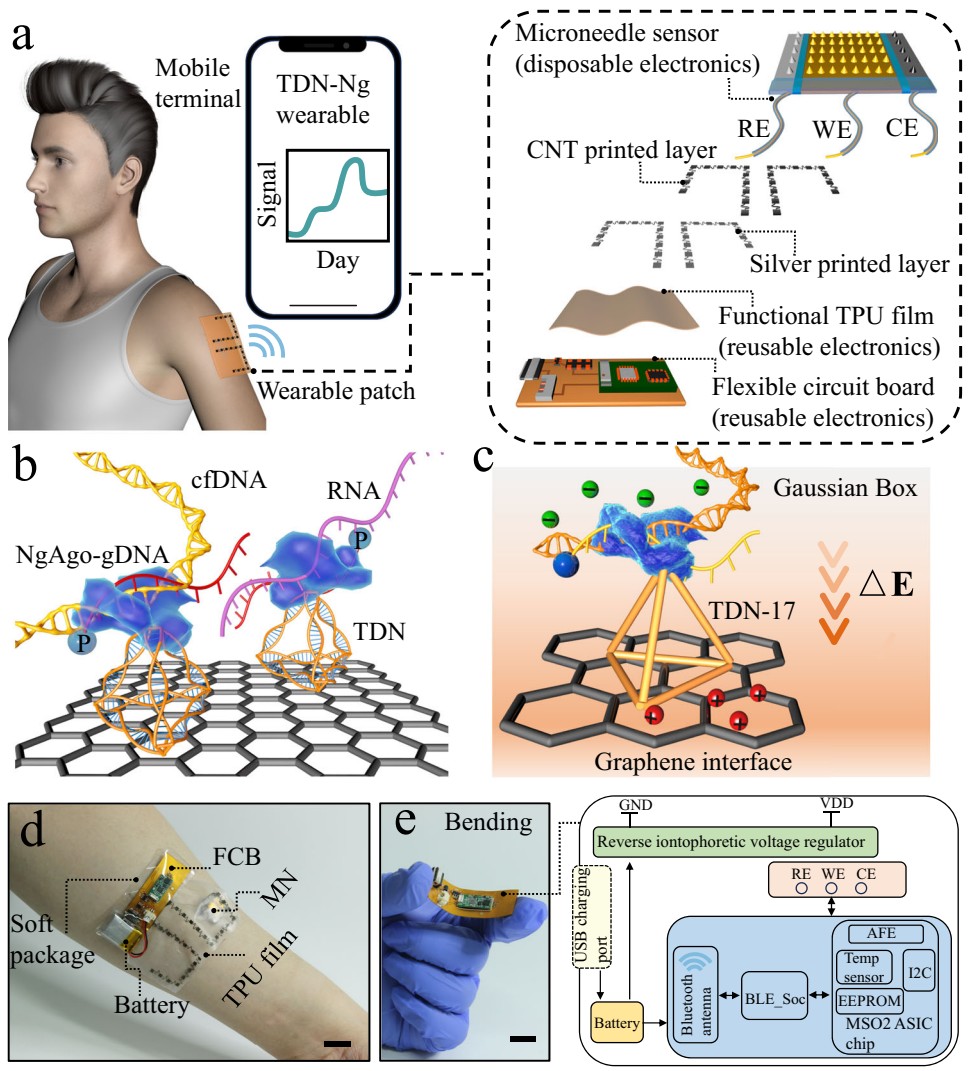

**Fig. 1 | Overview of the integrated wearable system based on TDN-Ng for real-time monitoring of nucleic acids. a** Workflow and view of the wearables on patients' epidermis for practical use. All the data were transmitted to an electronic device, RE, WE, and CE referring to the reference, working, and counter electrodes, respectively. **b** Illustration showing the engineered biosensing interface on a microneedle, in which TDN is used as an anchor to combine with graphene and NgAgo-gDNA, P referring to phosphorylated. **c** Schematic of the biosensing mechanism. The enhanced TDN-Ng interfaces selectively recognize target nucleic acids and produce different potentials based on Gaussian box theory, generating electrochemical signal outputs. **d** A photograph of the whole integrated wearable system, which was worn on the patient's arm. Scale bar, 1 cm. **e** Block diagram of the flexible circuit board electronics. The electrochemical analog front end (AFE) performs electrochemical sensing to acquire the signal, and the Bluetooth low-energy (BLE) module processes the signal and transmits the data to an outer electronic device via a Bluetooth antenna. The reverse iontophoresis module is individually connected with a battery. The battery was connected to the board and USB charging port. Scale bar, 1 cm.

nucleic acids via reverse iontophoresis. The third part of the integrated wearable system is the self-designed and customized FCB for power and signal management.

Figure 1b shows a schematic diagram of the MN interfacial structure. Here, TDN-17 was used as the optimal nano-scaffold to be immobilized on the surface of graphene. In this research, the enhancement of the sensitivity of the system may be derived from the spatial orientation of the TDN, as shown in Fig. 1c. The rigid and molecular-level mediation of TDN enables ultratrace biosensing in ISF, which largely involves an increase in the height of the electrical double layer. Atop TDN-17, the NgAgo protein, as one of the prokaryotic argonautes, was fixed to form the NgAgo/guide DNA complex, denoted as NgAgo-gDNA. Under the guidance of gDNA (24-bp 5′-phosphorylated single-stranded DNA), the entire target sequence is searched via Watson-Crick base pairing. Once matched, the immobilized NgAgo-gDNA could bind with the target nucleic acid, which is well-defined stable due to the candidate gDNA.

In this study, the as-fabricated MN was improved in three aspects: (I) As a hydrophobic, rigid and nonexpansive material, SU-8 photoresist was used to prepare MN by lithography, which is of great significance for the construction of high-performance biosensing interfaces (characterization in Supplementary Fig. 1); (II) as a nano-scaffold, TDN participates in the biosensing interface to generate enhanced signal output; and (III) NgAgo-gDNA was anchored at the top of TDN to form TDN-NgAgo-gDNA (denoted TDN-Ng), which improved the long-term stability of the assay due to the use of gDNA.

Figure 1d provides an overview of wearable electronics on the epidermis of a wearer, indicating the properties of full integration, skin conformation, and compactness. To acquire recorded electrochemical data and transmit it to a mobile terminal, wearable electronics use three circuits, including an analog front end (AFE), reverse iontophoresis and Bluetooth low-energy (BLE), as shown in Fig. 1e. The AFE offers a circuit for MN patch biosensing. The reverse iontophoresis module is independently powered by a battery to avoid signal crosstalk. The BLE, which performs as a data acquisition module, interacts with the AFE. All these subcomponents are powered by a rechargeable lithium-ion battery.

## Fabrication, material strategy, and evaluation of the wearable electronics

In this study, we further optimized the substrate material and fabrication method of MN patch. The MN patch as an actuator plays an important role in the wearable system, where interface stability is one of the significant factors for biosensing. Supplementary Fig. 45a shows the whole structure of the three-in-one MN patch, where working, reference, and counter electrodes were integrated into one MN patch for convenient assembly into FCB in real-world use. The bare MN patch was prepared via lithography using a hydrophobic rigid substrate SU-8 photoresist (Supplementary Fig. 2). Then, metalation, insulation, packaging, and surface biochemical modification were successively employed for the SU-8 MN patch (details in Methods).

From Supplementary Fig. 45b of the scanning electron microscopy (SEM) photograph and Supplementary Fig. 3, an off-the-shelf SU-8 MN patch had a well-defined shape, with a height of 800 μm ± 10 μm, a basic radius of 300 μm ± 10 μm, and a top radius of 15 ± 5 μm. In Supplementary Fig. 45c, d, the as-prepared SU-8 MN patch was applied to penetrate piglet skin and analysed by histology with haematoxylin and eosin (HE). The results showed that MNs were obviously inserted into the epidermis of the sample. Compared with our previously proposed hydrogel MN patch, the rigidity of the SU-8 MN patch was improved by ~2.4-fold, and it had the capability for electrochemical signal recording (Supplementary Fig. 4). Thus, we believe that the prepared SU-8 MNs might be a versatile platform for the field of wearable technology because they are superior to hydrogel MNs in terms of low time consumption, operability, stability, and rigidity.

To conformally laminate the epidermis of human skin, a flexible and conductive TPU film with a reverse iontophoresis compartment has been developed for wearable electronics. TPU-based wearables have been applied for human motion monitoring and e-skin due to their high gauge factor and working strain[29,30]. In this study, we optimized the preparation process of TPU films, and customized silver ink and carbon nanotube ink were successively spray-printed on the surface of the TPU film to decorate conductive patterns (Supplementary Figs. 5–8, supplementary movie 2). The representative functional TPU patch was verified by a finite element analysis (FEA) simulation under stretching, twisting, and bending mechanical distortions, as shown in Supplementary Fig. 46. The TPU patch had a lower theoretical modulus than that of our previously reported functional PDMS patch (maximum value of ~0.07 MPa), with a maximum value of ~0.016 MPa at 16% stretch, which is comparable to the human skin modulus[31], indicating its remarkable skin conformity and softness. Additionally, we used customized silver ink as the intermediate substrate to construct dual-layered patterning to enhance the adhesion and conductivity of the TPU patch. The SEM images of the Ag layer and double layer in Supplementary Fig. 46 illustrate that TPU was successfully decorated with dual-layered conductive patterning.

In Supplementary Fig. 47, mechanical testing was employed for the TPU patch to demonstrate its stretchability and elasticity. The maximum elongation of the functional TPU patch was at a break of 210% working strain in the range of ~0.9 MPa. During single stretch-release at 10% strain, the hysteresis phenomenon might be ascribed to the dual modification on the surface of TPU. Endurance tests confirmed that it had the ability in fatigue resistance to some extent, with a coefficient of variation (C.V.) of 6.9% in a 100-cycle measurement. Stretchable electronic devices have difficulty meeting the requirements of enhancing both the high gauge factor (GF) and working strain at the same time, which reflects the sensitivity and elasticity[32]. In this study, the GF value of the TPU patch reached 234.7 within a working strain of 210%.

Then, the spray-printed dual-layered morphology on the surface of the TPU film was directly verified by a stylus profiler, as shown in Supplementary Fig. 48a, indicating that the spray-printing material was evenly distributed on TPU with a height of ~8 μm (Supplementary Fig. 9). Both the patterned TPU film and PDMS film showed comparable optical transparency, as shown in Supplementary Fig. 48b. The biocompatibility of the MNs and TPU patches was explored by the MTT method, as shown in the Supplementary Information (Supplementary Fig. 10). All these results showed that the SU-8 MN patch could be used as an actuator of wearable biosensing systems in vitro or vivo, and the as-prepared TPU patch was probably a versatile platform for flexible electronics.

## Performance and mechanism of TDNs for surface biosensing

The strategy of interface construction is the key to improving the sensitivity of biosensing. In our previous study, we found that the electric double layer on the surface of graphene exerted an impact on the Donnan potential, which produced electrical signal output. Then, we tried to adjust the spatial height of the NgAgo-gDNA complex atop the graphene surface (Supplementary Figs. 11–12, Supplementary Notes 1). We utilized a double-stranded DNA ladder (10 bp to 70 bp, length from ~3 nm to ~21 nm). Similarly, we also found that the interface with a 20 bp ladder of double-stranded DNA (length of ~6 nm) had the maximum signal among these groups. Collectively, these data led us to hypothesize that the height of the NgAgo-gDNA complex modified on the graphene surface probably contributed to the generation of different electrical signals.

Hence, we further utilized a rigid and molecular-level probe, namely, TDN, to explore this hypothesis. In this study, TDN was used as the carrier of the molecular recognition complex on the surface of the graphene-modified MN patch. As far as we know, TDN, as one of the

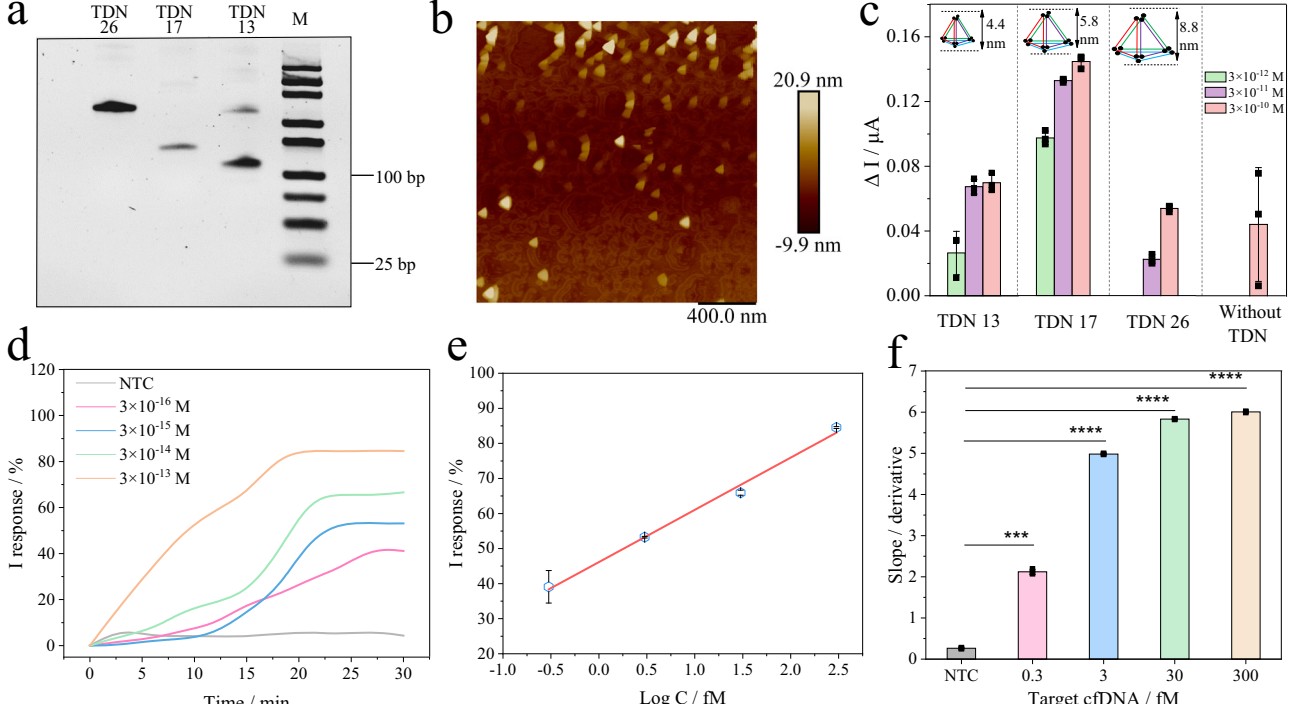

**Fig. 2 | Investigation of the impact of different TDN-modified biosensing interfaces. a** PAGE electrophoresis of TDN-13, TDN-17, TDN-26. 5% page, 110 V, 70 min, the whole process below 10 °C in ice-water bath, dyed for 15 min in room temperature, M referring to DNA marker, $n = 3$ independent experiments, repeated time=3 for each independent experiment. **b** AFM image of TDN-17, $n = 3$ independent experiments, repeated time = 3 for each independent experiment. **c** Comparison of the response signals in the presence of different concentrations of target cfDNA with TDNs of different heights and without TDNs in simulated ISF (PBS, pH = 7.4, 37 °C, 0.01 M), $n = 3$ independent experiments, repeated time=3, data presented as the mean values ± standard deviations (SD). **d** Dynamic curves of real-time monitoring target cfDNA by TDN-17 (height of 5.8 nm)-mediated biosensing interface on a microelectrode under reverse iontophoresis (10 V, PBS, pH = 7.4, 37 °C, 0.01 M). **e** Calibration curve of I response versus target concentration. Data are presented as the mean values ± SDs, $n = 3$ independent experiments. **f** Slope values of the curves collected from Fig. 2d calculated by differentiation using two-way ANOVA: *$p < 0.05$, **$p < 0.01$, ***$p < 0.001$, ****$p < 0.0001$, $p$ value of 0.000477, 0.00000598, 0.00000752, 0.00000842 for 0.3, 3, 30, 300 fM groups respectively, data presented as the mean values ± SDs, $n = 3$ independent experiments.

DNA framework nanostructures, has the advantages of efficient hybridization regulation, an ordered interface, programmability and high sensitivity[33]. More recently, TDNs with remarkable stiffness have been increasingly applied for biosensing a variety of targets, such as ions, proteins, RNA, DNA, and cells[27–29,34,35]. According to the developed protocol[26], we prepared functionalized TDNs using a one-pot annealing bottom-up method of four single-stranded DNAs and optimized the TDN-based interfaces for biosensing. Finally, TDN-17 was chosen as the optimal choice for the follow-up experiment. The sequences of TDNs are listed in Supplementary Table 1 (provided by Sangon corporation, Shanghai). The corresponding characterization of the as-prepared functionalized TDNs is shown, including transmission electron microscopy (TEM), SEM, atom force microscopy (AFM), and PAGE electrophoresis (Fig. 2a, b, Supplementary Fig. 13, 14). These data consistently supported the successful preparation of different TDNs.

Due to the low abundance of disease-related biomarkers in ISF or other biofluids (e.g., DNA, RNA, protein), sensitive monitoring of ultratrace targets in bulk samples is of great significance. To explore whether the as-prepared TDN-17 can improve the biosensing sensitivity, the engineered NgAgo-gDNA complex was first anchored atop a TDN-17-modified graphene surface. According to the characterized potential of the constructed biosensing system at -0.28 V, the current signal of the biosensor with TDN-17 was increased compared with that of the system without the TDN system (Fig. 2c). According to the results, as the target concentration increased, the signal continued to increase, which was due to the continuous increase in Donnan's potential. However, we found that as the height of the TDN continued

to increase after TDN-17, the signal output and detection sensitivity decreased. The optimal height was approximately 5.8 nm for TDN-17. These results preliminarily showed that the spatial height has a certain regulatory effect on Donnan's potential of the interface within the range of suitable heights. At the same time, we also discovered that, compared with the blank group without TDN, the detection limit of TDN-17 (5.8 nm height) was reduced by two orders of magnitude, reaching $3 \times 10^{-12}$ M. In Fig. 2d–f, we primarily explored the detection limit of the TDN-17/engineered NgAgo-gDNA on a chip (reverse iontophoresis of 10 V) by a commercial microelectrode (schematic in Supplementary Fig. 15 and note 2). Strikingly, the detection limit was reduced to $3 \times 10^{-16}$ M in comparison to the NTC group. This result is of great importance for subsequent experiments to investigate a higher sensitivity for MN patches.

It is known that two key factors affecting the sensitivity of the surface are the probe spacing (for electron transmission) and the effective quantity of the probe (the possibility of collision). The TDN can improve both factors that are in a comprehensive relationship. Based on these results, it is necessary to further explore the mechanism of TDN-mediated biosensing surfaces in improving sensitivity. The mechanism was related to the concept of a Gaussian box, which contains all the charge in the diffuse layer and tight layer (electrical double layers) based on the Gouy-Chapman-Stern model. Then, the potential in the electrical double layers largely depends on the Debye length, which measures the charge carried in solution as well as the continuous spatial range of the electrostatic effect. Within the Debye length, charged substances in electrical double layers output the

current response. If the electrical double layers exceed the Debye length, the biosensing signal output might be impaired. The detailed theory and deduction are discussed in the Supplementary Information (Supplementary Fig. 16 and Note 3).

## Design and validation of the engineered DNA-guided NgAgo system for nucleic acids

One of the critical aspects is whether the engineered DNA-guided NgAgo system (Ng system) can recognize and bind with target nucleic acids, which is of great significance for subsequent research. In some previous research[36–38], prokaryotic Argonaute (Ago) can be divided into short-length and full-length Ago (pAgo). Most pAgo consists of four domains: amino-terminal (N-terminal), Piwi Argonaute Zwille (PAZ), middle (MID) and P element-induced wimpy testis (PIWI). The structure of those pAgo forms double lobes, where one lobe consists of the N-terminus and PAZ and the other consists of MID as well as PIWI. The target nucleic acid recognition or cleavage is guided by a 5′-phosphorylated oligonucleotide guide strand including DNA[39]. A binding pocket consisting of MID and PAZ promotes the spatial anchoring of the guide strand[40], where MID interacts with the 5-end of the guide to facilitate its binding to the target nucleic acid[41], while PAZ interacts with the 3-end of the guide to avoid its degradation[42]. Recent advances in pAgo-based biotechnology have led to the development of a flexible platform for genome editing and programmable nucleic acid biosensing.

Most studies have focused on the application of hyperthermophilic Ago for nucleic acid biosensing, such as pfAgo[43] and TtAgo[44]. There is a lack of mesophilic Ago with thermostability at a closer human body temperature, which is worthy of proposing as a wearable-based nucleic acid monitoring tool. Among these pAgo, one candidate is archaeon NgAgo, which has been debated regarding its DNA cleavage ability in eukaryotic systems. In addition to the four domains mentioned above, NgAgo was confirmed to have a fifth domain, that is, the RepA domain[39]. More recently, it was reported that NgAgo not only might bind with target DNA via the RepA domain[39] but also knockdown the fabp11a gene and block transcription through binding to the coding sequence[45,46]. Similar to other pAgo systems, a 5′-phosphorylated 24-nt single-stranded oligonucleotide as the guide DNA (gDNA) is essential to form the NgAgo-gDNA complex. Based on the principle of other pAgo technology, it was speculated that the complex might search the entire nucleic acid sequence under the guidance of gDNA, where a 5′-phosphorylated 24-nt specific sequence matches the target. Once matched, the complex can specifically bind with the target directly. In this study, we revisited this protein and tried to use only its binding ability to accommodate wearable biosensing actuation modules, without regard to its potential in genome editing or DNA cleavage.

To verify the feasibility of the Ng system, we attempted to test the recognition of nucleic acids by NgAgo-gDNA in vitro. First, the engineered NgAgo expression, mass spectrometry identification, and SDS-PAGE diagram are shown in Supplementary Fig. 17. We constructed the Ng system and screened its optimal guide DNA 3 for subsequent experiments (Supplementary Fig. 18, 19, oligo sequences provided by Sangon corporation Shanghai, in Supplementary Table 1). In Supplementary Fig. 44, from the electrophoretic mobility shift assay (EMSA), the bands of binding events had an obvious drag shift. Additionally, through the EMSA, the band of the Ng reaction had a drag shift, which is analogous to CRISPR-dCas9, because the FAM-labeled target DNA had a slower mobility with the binding of proteins. The above results primarily confirmed the recognition between the NgAgo-gDNA complex and target DNA.

Then, we tested its binding affinity via surface plasmon resonance (SPR). The SPR chip construction, sample preparation, and method are discussed in Supplementary Fig. 20 and the Methods section. From the results of Fig. 3a–c, in the range of 3.125 nM–100 nM of target EBV cfDNA, there was a certain linear relationship between the relative response and concentration of the NgAgo-gDNA complex, with a linear equation of $R_{max}$ (a.u.) = 0.6942·C(nM)+3.9867 (R = 0.9859). According to the affinity constant equation, strong binding between the NgAgo-gDNA complex and EBV cfDNA was determined, with a $K_D$ of $5.49 \times 10^{-9}$ $M^{-1}$, while the $K_D$ of CRISPR-dCas9 was $1 \times 10^{-9}$ $M^{-1}$ according to a previous report[47]. The targeting affinity of the Ng system might be slightly stronger than that of CRISPR-dCas9.

To eliminate the interference of the graphene surface, we constructed a simple electrochemical biosensor to further investigate the binding affinity (processes in the Supplementary Methods). In Fig. 3d, compared with the NTC group, the gDNA 3 group had an obvious signal response with a significant difference. Additionally, we found that the gDNA 3 group had the maximum current irrespective of the graphene interface, which is consistent with the results mentioned above.

Theoretically, the predicted NgAgo-gDNA model was shown in Fig. 3e. The NgAgo protein was expected to have a crystal structure consisting of canonical N-terminal, PAZ, MID, PIWI, and RepA domains. gDNA was stably bound with a pocket formed by the N-terminus, PAZ, MID, and PIWI. The charge distribution of the NgAgo-gDNA complex was also visualized from Fig. 3e(iii)–(v), showcasing the negatively charged surface in a large area, which was consistent with the mechanism of electrical double layers mentioned above. It was observed that the surface of the RepA domain colored with positive charges was suitable for interaction with negatively charged substrates, such as DNA or RNA, in accordance with a previous report[39]. During the molecular dynamics simulation, the NgAgo-gDNA complex was highly stable via hydrogen bonds. A detailed discussion was shown in Supplementary Fig. 21, Note 4, supplementary movie 1.

According to the above results, it is reasonable to believe that the engineered Ng system could effectively bind and recognize target EBV cfDNA. Furthermore, we intended to verify this fact by a series of methods, including electrochemical biosensors and UV-vis spectroscopy (Supplementary Figs. 22–24 and Note 5). In this study, the engineered Ng system exerted an advantageous effect on wearable biosensing: (1) NgAgo is predictably smaller size than that of CRISRP effector (e.g., Cas9, Cas12a)[36], which is efficient for coupling with TDN-17 to avoid overlap, noncovalent binding, and aggregation on the surface of graphene; (2) it requires a DNA guide (usually from 15-24 nt length) without a protospacer adjacent motif, which might be more stable than sgRNA in vivo for wearable electronics; and (3) compared with the hybridization chain reaction method for biosensing DNA targets, the engineered Ng system had an improved signal response over 10-fold (Supplementary Fig. 25).

Hereby, in this research, we only unraveled the binding function of the engineered Ng system, and we still do not have full insight into how the NgAgo-gDNA complex binds to target cfDNA. Is the binding mechanism dependent on the allosteric NgAgo protein targeting to unwinding nucleic acid? Or does NgAgo conformation manipulate its targets? Undoubtedly, future research on its structure and biochemical properties will reveal these unknown questions.

## Evaluation of TDN-Ng MNs in vitro

As an integrated wearable electronics, the proposed TDN-Ng wearable system was ultimately supposed to realize real-time monitoring of target nucleic acids in a real scenario. To facilitate the practical applications of this system, we further investigated its sensitivity, quantitative detection, long-term stability, specificity, and anti-interference properties.

The as-fabricated TDN-Ng MNs can monitor different concentrations of target cfDNA in real time quantitatively and sensitively in vitro. In Fig. 4a, target cfDNA was continuously recognized and bound by the engineered Ng system on the MN surface in comparison to the NTC group. The concentration of target cfDNA was positively correlated

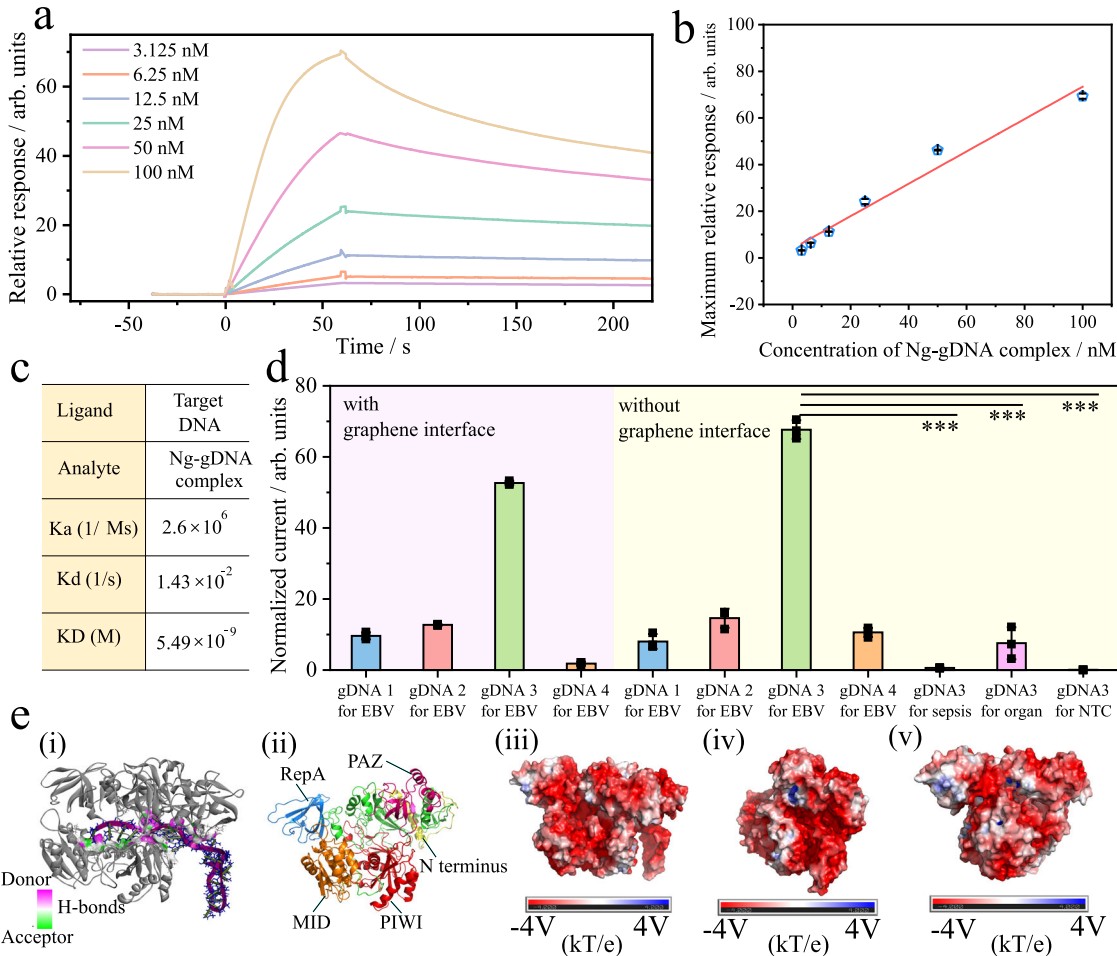

**Fig. 3 | Engineered NgAgo-gDNA binds to target nucleic acids with high affinity.** **a** SPR dynamic curves of the interaction between target DNA and NgAgo-gDNA complex of different concentrations. **b** Calibration curves of SPR, data presented as the mean values ± SDs, $n = 3$ independent experiments. **c** The affinity constant of target DNA and the NgAgo-gDNA complex validated by the SPR method. **d** The recognition ability of different gDNA-guided NgAgo systems to target DNA with and without a graphene surface; pink and yellow backgrounds refer to the graphene interface and no graphene interface, respectively, using 50 mM [Fe(CN)$_6$]$^{3-/4-}$. For the graphene interface group, NgAgo was immobilized on this surface, and 1% BSA blocked the nonactive sites. gDNA was then anchored by NgAgo. For the no graphene interface group, thioglycolic acid was first immobilized on the surface of the gold microelectrode, and the following process was the same as that for the graphene interface group, using two-way ANOVA: *$p < 0.05$, **$p < 0.01$, ***$p < 0.001$, ****$p < 0.0001$, $p$ value of 0.000511, 0.000322, 0.000479, for sepsis, organ, NTC groups respectively, data presented as the mean values ± SDs, $n = 3$ independent experiments. **e** The predicted modeling of NgAgo-gDNA (i) and the indicated domain of NgAgo protein (ii) containing RepA, N-terminus, PAZ, MID, PIWI, the color-coded surface charge distribution of the complex, (iii) to (v) referring to front, right, left view, respectively.

with the relative I response, with a linear equation of I response (%) = 7.3911·logC(fM)+40.9961, ($R = 0.9881$) in the range of $3 \times 10^{-16}$ M to $3 \times 10^{-13}$ M of target and with a detection limit of $3 \times 10^{-16}$ M (Fig. 4b). To verify the truly positive result of Fig. 4a, the differentiation of slopes was calculated for five groups via Origin software, as shown in Fig. 4c. It represented significant differences between target cfDNA groups and the NTC, and the slopes were positively proportional to the target cfDNA concentrations. Interestingly, it was shown that the plateau of the I response gradually tended to be stable within 30 min, which had a shorter monitoring time than our previously reported CRISPR MN patch (75 min), indicating the much faster reaction kinetics of our tetrahedral nanostructure-based NgAgo-gDNA system (detailed mechanism discussed in Supplementary Note 6). The platform could be employed for endpoint quantitative detection of EBV cfDNA, as shown in Supplementary Fig. 26 and Note 7.

The TDN-Ng MN platform with the biosensing element of the engineered Ng system can achieve long-term stability, where single-stranded guide DNA (gDNA) is needed instead of sgRNA. Unlike traditional lab-on-tube assays, wearables are inevitably exposed to harsh and uncontrolled environments and are in direct contact with the

human epidermis. To examine its long-term stability for the detection of target nucleic acids, the as-prepared TDN-Ng MN patch was kept in stimulated ISF (37 °C, pH 7.4) and used to monitor $3 \times 10^{-14}$ M of target every day under reverse iontophoresis (10 V). As shown in Fig. 4d, as time elapsed, the signal threshold ($S_t$, defined as the signal platform threshold) tended to decrease over 20 days. During the 16-day period, the $S_t$ values remained stable, with a C.V. of 4.98%. Although the $S_t$ values decreased with a C.V. of >5% from the 17-day period, obvious "S"-shaped signal curves were still observed. Collectively, these results indicated that the as-fabricated TDN-Ng MN patch exhibited a long-term stability of 16 days within reasonable C.V. values in vitro.

Meanwhile, from the data of Supplementary Fig. 27, we found that the time threshold value ($T_m$ defined as the time relative to the max of the signal response derivative) is independent of the time change. However, the $T_m$ values still had the capacity to distinguish between the positive group and the NTC group. The $S_t$ value rapidly declined after 20 days, without "S"-shaped curves. Given that the $S_t$ value depended solely on the binding activity of NgAgo-gDNA, the reaction rate of the TDN-Ng system may be close to the Michaelis-Menten law of enzyme kinetics.

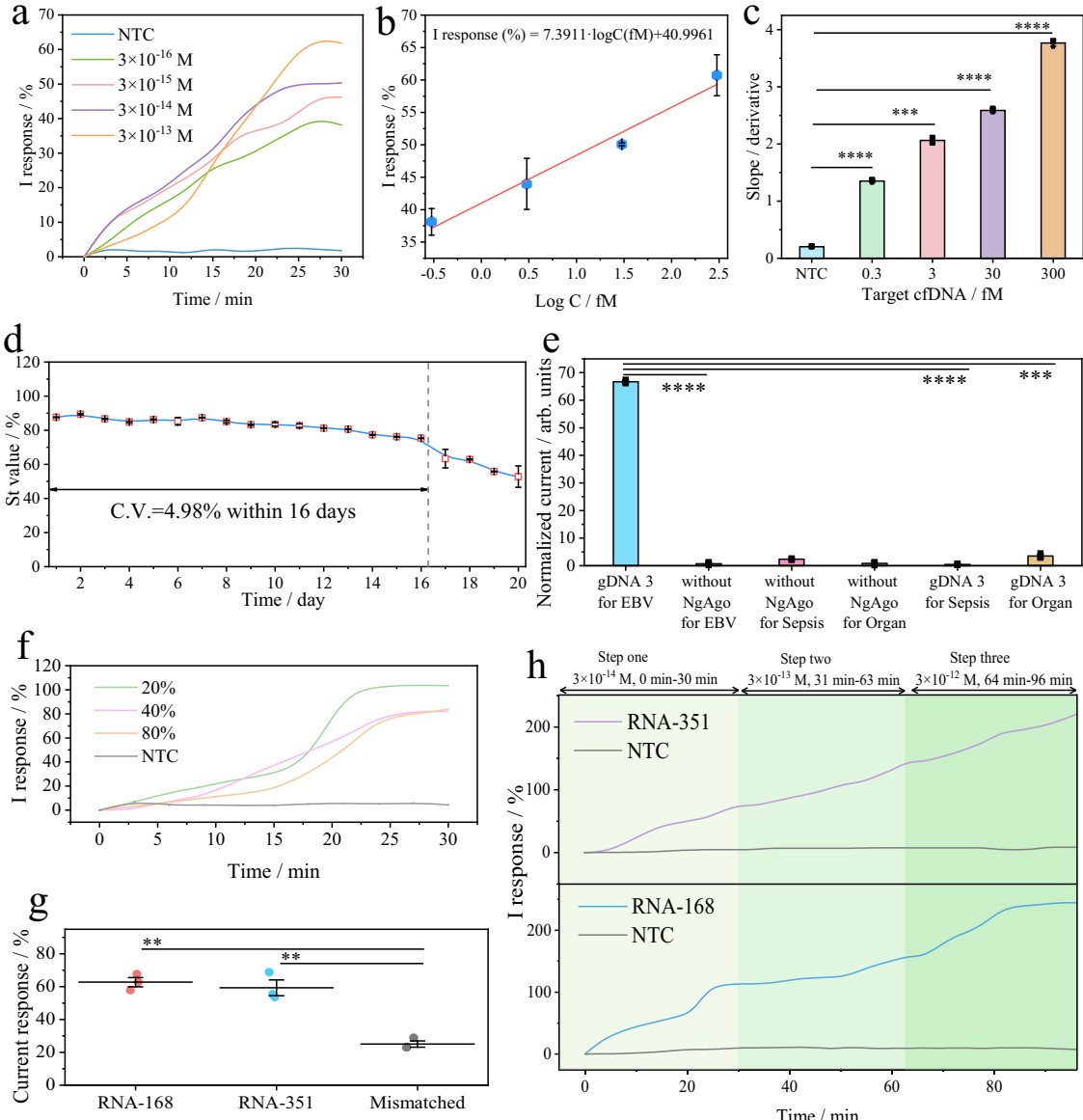

**Fig. 4 | Performance of TDN-Ng MNs for cfDNA and RNA in vitro. a** Real-time I response targeting variable concentrations of target cfDNA by the TDN-Ng MN patch in stimulated ISF, 37 °C, pH 7.4, and 0.01 M PBS. **b** Calibration curve of I response versus target concentration. Data are presented as the mean values ± SDs, $n = 3$ independent experiments. **c** Slope values of the curves collected from Fig. 4b calculated by differentiation, using two-way ANOVA: $*p < 0.05$, $**p < 0.01$, $***p < 0.001$, $****p < 0.0001$, $p$ value of 0.0000678, 0.000232, 0.000026, 0.0000483 for 0.3, 3, 30, 300 fM groups respectively, data presented as the mean values ± SDs, $n = 3$ independent experiments. **d** The long-term stability of the TDN-Ng MN patch in vitro for 20 days, incubated in stimulated tissue at 37 °C, and monitoring $3 \times 10^{-14}$ M target cfDNA on a chip under reverse iontophoresis (10 V). Data are presented as the mean values ± SDs, $n = 3$ independent experiments, by polynomial fitting of 4 orders. **e** Specificity of the TDN-Ng platform for 0.3 nM target and other interfering nucleic acids using 50 mM $[Fe(CN)_6]^{3-/4-}$, using two-way

ANOVA: $*p < 0.05$, $**p < 0.01$, $***p < 0.001$, $****p < 0.0001$, $p$ value of 0.00001, 0.000048, 0.000173 referring to, without NgAgo for EBV, gDNA 3 for sepsis, gDNA 3 for organ groups respectively, data presented as the mean values ± SDs, $n = 3$ independent experiments. **f** The anti-interference performance of the TDN-Ng MN patch under different concentrations of fetal bovine serum for $3 \times 10^{-12}$ M target cfDNA. **g** Investigation of the binding ability of the NgAgo-gDNA complex with 3 nM RNA-168 and RNA-351, incubated at 37 °C for 60 min, analysed by two-way ANOVA: $*p < 0.05$, $**p < 0.01$, $***p < 0.001$, $****p < 0.0001$, $p$ value of 0.0083, 0.0071, for RNA-168, RNA-351 groups respectively, data presented as the mean values ± SDs, $n = 3$ independent experiments, repeated time = 3. **h** Dynamic change in RNA-168 and RNA-351 recording by a TDN-Ng MN; steps 1, 2, and 3 refer to real-time monitoring of the $3 \times 10^{-14}$ M target during 0–30 min, the $3 \times 10^{-13}$ M target during 31–63 min, and the $3 \times 10^{-12}$ M target during 64–96 min, respectively.

The TDN-Ng MN patch can withstand 0.3 nM nonspecific nucleic acid and 80% fetal bovine serum (FBS) interference. In Fig. 4e, the system showed satisfactory selective detection of target EBV cfDNA in the interference of others, including sepsis cfDNA and kidney transplantation cfDNA (denoted as organ). Notably, compared with the no NgAgo groups, the positive groups showed obvious signal output, indicating that the NgAgo system exerted an important impact on the molecular recognition of target nucleic acids. Additionally, the stable

selectivity detection was further investigated in vitro and in vivo by cross-over gDNA sequences and animal models respectively, showing that the target DNA was specifically and evidently bound with NgAgo-gDNA system (Supplementary Fig. 40, 42, 43). The anti-interference ability of the proposed system was also examined for monitoring target cfDNA in the range of FBS concentrations from 20 to 80%. As shown in Fig. 4f, the current response is stable under 20% FBS with a C.V. of 5.04%. Additionally, it was observed that the groups with 40%

and 80% FBS interference still showed obvious "S" shaped curves within 30 min, with C.V. values of 18.63% and 19.16%, respectively. Compared with the total protein concentration of human body ISF (20.6 mg/mL)[48], this excellent anti-interference ability to 80% FBS (23 mg/mL) might allow TDN-Ng MN wearable patches to be applied in real applications.

According to previous research[45,46], the NgAgo-gDNA complex has been reported to have the ability to knock down genes and induce transcriptional silencing. Thus, we suspect that RNA might also be recognized by our TDN-Ng system. We first screened RNA-168 and RNA-351 sequences as disease-associated biomarkers (provided by Sangon corporation Shanghai, Supplementary Table 1) and then used engineered NgAgo-gDNA to identify these two RNAs in ISF. In Fig. 4g, compared to the control group, RNA-168 and RNA-351 bound to the NgAgo-gDNA complex and induced a significant current response. In this study, we selected RNA-168 and RNA-351 as targets to verify the real-time monitoring capability of the TDN-Ng MN patch. In Fig. 4h, during step one of the $3 \times 10^{-14}$ M target, as the concentration of RNA-168 or RNA-351 increased, the signal increased. The signal continuously increased in successive steps two ($3 \times 10^{-13}$ M) and three ($3 \times 10^{-12}$ M), implying that this system was able to monitor the dynamic change in target RNA in real time. And the TDN-Ng MN platform could also real-time monitor other applications, such as staphylococcus aureus and pseudomonas aeruginosa in Supplementary Fig. 41. These data indicated that the TDN-Ng MN platform might be available to monitor longitudinal nucleic acids, including DNA and RNA.

## Performance and evaluation of in vivo monitoring with integrated MNs

In this work, we further validated TDN-Ng MNs in vivo for real-time monitoring of nucleic acids. The validation workflow is shown in Fig. 5a. First, an EBV-mouse model was constructed and applied to demonstrate the feasibility and reliability of the wearable platform for multiple real-time monitoring in vivo, including cfDNA and RNA. Accordingly[49,50], EBV-related nucleic acid biomarkers have been reported to be associated with sepsis, and it has been reported that cfDNA or RNA is present in CNE cell lines. Thus, we first reprogrammed CNE cell lines with the luciferase reporter gene (Luc) and sub-cutaneously inoculated them into 8-week-old female BALB/c nude mice. The corresponding primer screening, calibration curves, RT-PCR for ex vivo and in vivo CNE cells, gene typing, and next-generation sequencing (NGS) are discussed in detail in Supplementary Figs. 28–32, Note 8, and Table 1. The complete cellular and animal procedures are described in the Methods section. The TDN-Ng MNs were applied to BALB/c nude mice under reverse iontophoresis components (10 V). To avoid signal crossing, the integrated MN adopted an intermittent working mode, as shown in Fig. 5b. It is worth noting that, due to the long-term stability of the MNs in our study, only one MN patch was used for one animal during the whole process of 7 days. The working mode is as follows: (1) Before each use, the MN patch was calibrated for 100 s, (2) reverse iontophoresis was performed for 3 min to pre-enrich the target, and (3) the MN biosensor started to collect electrochemical data for 100 s. Steps 2 and 3 were repeated until the end of the experiment.

As shown in Fig. 5b, we demonstrate parallel trials to monitor EBV cfDNA in vivo. Gold standard PCR was conducted to verify the accuracy of the TDN-Ng MN in this work. A commercial Whatman card was used for mouse ISF sampling (Supplementary Fig. 33), and the nucleic acid in the sample was extracted by a DNA extraction kit, which was then transferred to PCR. The signals of the TDN-Ng MNs (I response of 88.26%), PCR (Ct value of 27.35) and bioimaging (maximum of 2800 a.u.) were significantly the same at the 4 h timepoint. Then, at the 24 h time point (I response of 85.42%), the EBV cfDNA signal in mice monitored by the TDN-Ng MN method was slightly lower than that at the 4 h timepoint. Meanwhile, the PCR data (Ct

value of 27.41) and bioimaging signal (maximum of 1100 a.u.) changed slightly, which was consistent with our proposed method. Continuously, at the 48 h timepoint, the TDN-Ng MN signal decreased to 80.6%, which was the lowest among the other timepoint groups. A similar phenomenon was also observed in the PCR data (Ct value of 27.66) and bioimaging data (716 a.u.). Then, from the 168 h data, the TDN-Ng MN still had long-term stable sensitivity for monitoring EBV cfDNA in vivo, with a response of 107.24%, and PCR (Ct value of 26) as well as bioimaging (maximum of 4800 a.u.) also peaked. Additionally, CNE cells progressed and accumulated in mice (white dashed circle). At the same time, the TDN-Ng MN system was also available for the real-time monitoring of target RNA in EBV-model mice in vivo (Supplementary Fig. 34). The TDN-Ng MN patch had a correlative trend to the gold standard RT-PCR. Additionally, the concentrations quantified by the MN patch were 1.02 nM and 0.18 nM for cfDNA and RNA, respectively, at the 24-hour time point. Above all, this TDN-Ng MN system could not only effectively distinguish positive and NTC groups but also reliably monitor the dynamic changes in target cfDNA and RNAs in vivo.

In Fig. 5c, the real-time I response plot was intended to show that the disease group was true positive when compared to the healthy group (data from Fig. 5b). There were indeed significant differences between the positive and NTC groups by differentiating real-time signal plots. We also compared the $S_t$ value of the I response plots with the Ct value obtained from the standard PCR method at various timepoints, as shown in Fig. 5d. Consequently, the I response was proportional to the Ct value, indicating that the TDN-Ng MN system exhibited reliability in recording dynamic changes in target cfDNA in vivo with the same trend as the PCR method.

The calibrated cfDNA and RNA levels recorded by TDN-Ng MN closely tracked those of the standard PCR method (sampling by Whatman card and extraction nucleic acid by commercial kit), with a transfer time lag (~3 min). In Fig. 5e, the Pearson coefficients (denoted as Pearson's r) for the two sets of data were found to be 0.99 and 0.83 for cfDNA and RNA, respectively, demonstrating the reliable performance of wearable TDN-Ng MNs in monitoring dynamic target DNA or RNA fluctuations in vivo in a multiple, accurate, quantitative manner.

In addition, this TDN-MN device was further investigated for its long-term stability in the sensitive detection of target nucleic acids in vivo. From the data (Fig. 5f, Supplementary Fig. 35), it could be observed that the TDN-Ng MN remained stable over a 14-day time course, with a C.V. of 4.87%, where the $S_t$ value was in the range of 56.35–48.13%, while the $T_m$ value was in the range from 2.95 to 21.21. It still showed an obvious signal response and time threshold on day 17. Due to the rigid TDN structure and DNA-guided NgAgo system, the proposed MN was able to tolerate a complicated microenvironment in vivo. Accordingly, it has been reported that the median duration of ICU stay for sepsis patients is 11 days[50]. Thus, to the best of our knowledge, the 14-day reliable stability of the TDN-Ng MN device for the real-time monitoring of nucleic acids in vivo should be satisfactory and meet the clinical requirements for ICU patients, which might be the most satisfactory results for in vivo long-term stable monitoring of biomacromolecules.

To further examine the ability of detecting sepsis animal models directly, the TDN-Ng patch with specific gDNA was able to monitoring two kinds of sepsis mice bearing staphylococcus aureus and pseudomonas aeruginosa (denoted as SA and PA, respectively), within the period of inflammatory cytokine storm. Shown in Fig. 5g, h, the peaks of signal responses were 4-hour time point and 12-hour time point for SA and PA sepsis mice, respectively. And the TDN-Ng MN patch had a correlative trend to gold standard PCR (Supplementary Figs. 38, 39). These results showcased that the TDN-Ng MN patch was able to real-time monitoring longitudinal DNA within the period of inflammatory cytokine storm for sepsis animal models.

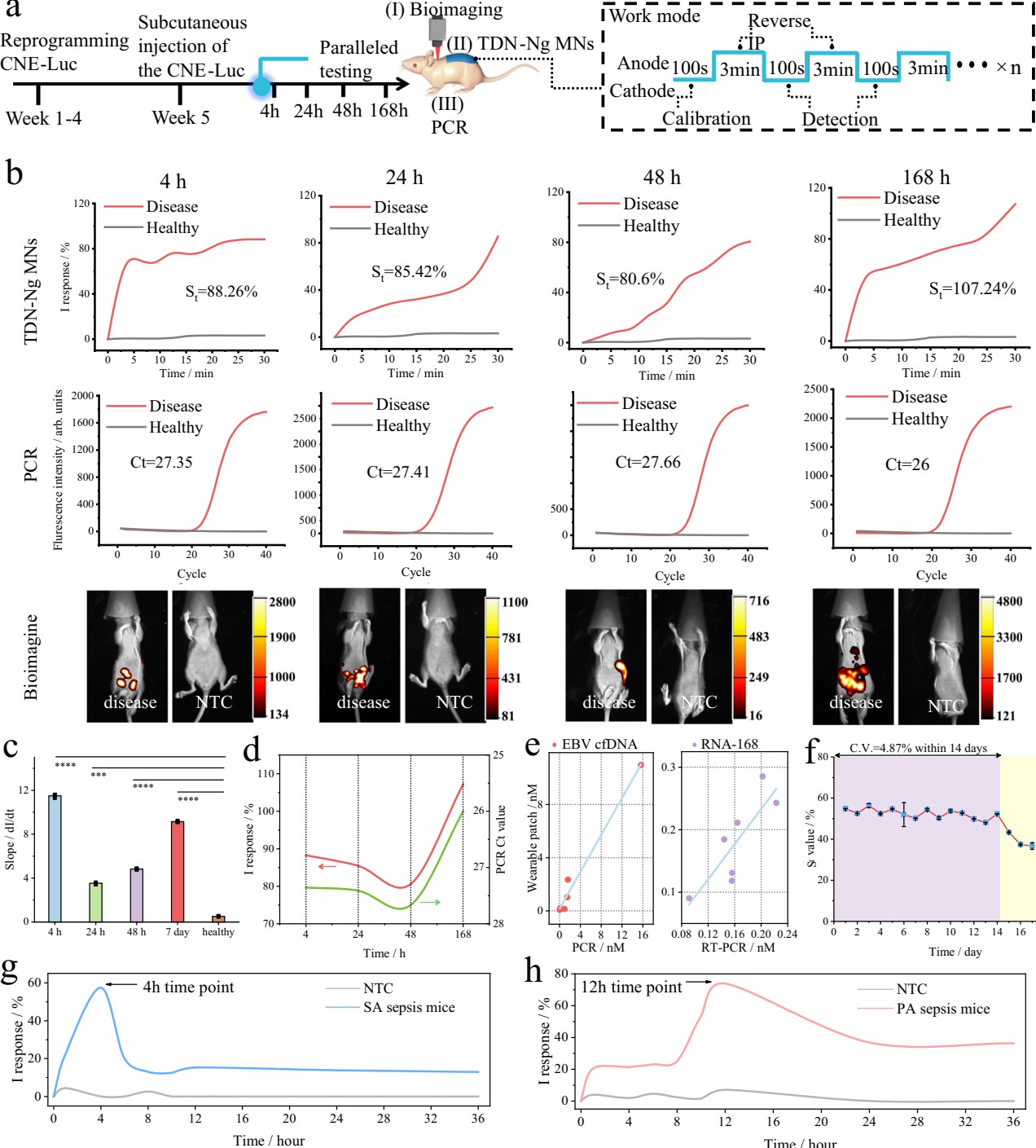

**Fig. 5 | In vivo verification of the TDN-Ng MNs. a** Schematic and timeline of real-time monitoring in the model mouse experiment, one TDN-Ng MN patch for one mouse throughout the experiment. each mouse undergoing (I) chemiluminescence bioimaging, (II) TDN-Ng MNs, (III) PCR assay. The TDN-Ng MN patch was first calibrated in PBS (37 °C, 0.01 M, pH 7.4) for 100 s to eliminate sensor background variation before use. The blue line represents the work mode. The skin of the mice was treated by the following steps: stratum corneum scrub cream, disinfection with 75% ethanol, smearing with TE buffer (pH 8.0), and drying with cotton. Mice were fixed on a heat plate throughout the process. **b** Parallel demonstrations of mice at different time points, including 2 h, 24 h, 48 h, and 168 h. **c** Slope values of the real-time dynamic curve at 2 h, 24 h, 48 h, 168 h, *p < 0.05, **p < 0.01, ***p < 0.001, ****p < 0.0001, analysed by two-way ANOVA, p value of 0.000072, 0.00028,

0.000022, 0.0000053 for 2 h, 24 h, 48 h, 168 h, respectively, data presented as the mean values ± SDs, n = 3 biologically independent animals. **d** Dynamic changes in target cfDNA in vivo recorded by TDN-Ng MNs and gold-standard PCR. **e** Corresponding plots of target cfDNA and RNA measurements by TDN-Ng MNs and gold-standard PCR and RT-PCR. **f** The long-term stability of the TDN-Ng MNs in vivo. The TDN-Ng MNs were laminated on the epidermis (37 °C) when not used and then applied for monitoring $3 \times 10^{-14}$ M target cfDNA under reverse iontophoresis (10 V). Data were recorded as the mean values ± SDs, n = 3 independent experiments. **g** Continuous real-time monitoring of SA sepsis mice within 36 h by TDN-Ng MN patch. **h** Continuous real-time monitoring of PA sepsis mice within 36 h by TDN-Ng MN patch.

## Discussion

While there has been great improvement in wearable devices over the years according to Moore's law, the trend of state-of-the-art wearable technology is increasingly towards continuous and long-term monitoring of disease-derived macromolecular biomarkers (e.g., protein, hormones, nucleic acid, tumor mass)[21,51–53], in addition to those already reported small molecules or ions. Real-time monitoring of macromolecular biomarkers may provide insights into individual physiological conditions, especially for early-stage diagnosis, alarm management, prognosis, and pandemics such as SARS-CoV-2[15,51]. For instance, as a global health problem, sepsis causes approximately one-third to a half of deaths in hospitals and contributes to ~11 million deaths worldwide every year[54]. Sepsis is defined as a host's dysregulated response to infection resulting in organ dysfunction[55]. Most sepsis patients with life-threatening organ dysfunction are directly transferred to the ICU[55]. In a statistic[49,50], it was reported that EBV-associated nucleic acids might be an alternative biomarker to predict clinical deterioration for sepsis patients in the ICU. Some ICU staff complained that more sensor cables hindered ICU patient health management[56]. It's important to increase frequency of monitoring sepsis patients in ICU[55]. The total length of ICU stay was approximately 11 days[50], and our proposed TDN-Ng wearable device has a stable sensitivity of 17 days in vivo, which could be competent to achieve continuous monitoring for ICU sepsis patients, in comparison to the commercial continuous glucose monitor of Freestyle Libre (Abbott Inc.) with a 14-day lifetime. It is necessary to develop wireless, real-time monitoring, non-invasive or minimally invasive, remote controlling sensors in ICU scenarios, either to shoulder the burden of the medical system resulting from demographic change or to ameliorate ICU patient alarm management.

Although biosensors, such as solid microelectrodes, microfluidic chips, and field-effect transistors, maintain high sensitivity (~$10^{-9}$ M to $10^{-16}$ M)[57–59], they are not wearable and cannot achieve real-time on-body sensing. More recently, microneedles have been divided into two types in terms of practical applications: drug delivery[60,61] and diagnostics[62–64]. Microneedle patches play a vital role in wearable biorecognition elements, where the interfaces can be reprogrammed by various nano-scaffolds to enhance signal output. However, microneedles are also confronted with technical barriers, such as sensitivity and stability.

The TDN-Ng wearable addressed these barriers and enabled real-time monitoring of longitudinal biomarkers. First, the rigid nanoscale TDN scaffold was more suitable for orientating the anchoring Ng protein, not only to improve the recognition resolution but also to eliminate steric hindrance. The TDN scaffold manipulates the geometry changes of the biosensing interface, such as height, ordered arrangement, and biorecognition element density. These efforts of the TDN scaffold allowed for high sensitivity. Second, the NgAgo protein was utilized to bind with target-specific nucleic acids, where a DNA oligo as a guide sequence is more stable than sgRNA in harsh microenvironment[65,66]. Although it remains the subject of intense debate regarding its gene cleavage in eukaryotic systems, some research showed that Ng exhibited the ability to inhibit gene expression via specifically binding to targets. They verified that the NgAgo-gDNA system offered a method for gene knockdown, similar to the dCas9/sgRNA system. Hereby, in this study, the NgAgo protein did nothing but affinity binding. The other gene-editing functions are unknown. Collectively, these innovations pave the way for amplification-free real-time monitoring. In Supplementary Table 2, 3, in comparison with other state-of-the-art amplification-free strategies and MN patches, our proposed TDN-Ng wearable device showed advantages in long-term stable sensitivity.

In summary, this study proposed wireless, integrated wearable electronics for real-time monitoring of longitudinal nucleic acids for sepsis-related animal models through engineering biosensing interfaces. Based on the synergetic effect of the TDN and Ng systems, the wearable system was able to continuously track dynamic changes in cell-free DNA and RNA targets in vivo, with a sensitivity of 0.3 fM and reliable stability within 17 days in vivo. We also discussed the mechanism of sensitivity enhancement and real-time monitoring dynamic equilibrium in the TDN-Ng system caused by TDN-17 with spatial height. The amplification-free strategy of this current work represented a leap forward for nucleic acid-based wearables and accelerated the emergence of next-generation biosensors for ICU health care management. The following development of this work will focus on directly recording fluctuating input signal (increase, decrease, and recovery), higher affinity construction, biosensing membrane regeneration and automated data acquisition. Meanwhile, artificial intelligence-based machine learning could be used to assist in the prediction of data acquisition for ICU health care alarm management.

## Methods

### Ethical Statement

The research guides the Guidelines of Laboratory Animals of Fudan University and complies with all relevant ethical regulations (Animal Ethics Committee of Fudan University, approval NO. 2021JSCHEM-020). And human research was approved by human Ethics Committee of Fudan University (approval NO. FE22187I). The informed consent has been obtained from all participants.

### Design, expression, and verification of Natronobacterium gregoryi protein

The expression and purification of Natronobacterium gregoryi protein (Addgene NO. 78253) was conducted by novoprotein incorporation (Shanghai). The verification of the protein was in Supplementary Fig. 17.

### Synthesis and purification of TDN

The sequences of different TDNs were listed in Supplementary Table 1 and purchased from Sangon Biotech Corporation (Shanghai). As for synthesis of TDN, firstly, 2 μL of four oligos (100 μM of stock solution) to form TDN was added into the centrifugal tube and 92 μL of TM buffer (purchased from Tansoole corporation, Shanghai) was added into this tube. Then the tube was rapidly centrifuged and vortexed to well mix all the oligos and buffer. Then, the tube was incubated in water-bath in 95 °C for 30 min. Finally, the tube was taken out for rapid centrifugation and vortex within 1 min, and immediately incubated in 4 °C for 45 min. The synthesized TDN solution could be stored in −20 °C for one week.

As for purification of TDN, the TDN samples was loaded into 5% PAGE for electrophoresis (110 V, 70 min), where the whole process was controlled under 10 °C using ice-water bath, and dyed for 15 min under room temperature. The PAGE was exposed and cut on the gel cutter (MiniPro™, ES-BL470, Yeasen, Shanghai) for selecting target bands. Then, the selected PAGE gel slices were conducted by PAGE DNA extraction kit (Sangon, Shanghai). Finally, the purified samples were verified by 5% PAGE electrophoresis (110 V, 70 min) again.

TDN was characterized by TEM (80 kV, HT7800, Hitachi, Japan), FESEM (Zeiss Gemini SEM500, Germany), AFM (multimode 8, Bruker, USA).

### Preparation, modification, and package of SU-8 microneedles

We designed the microneedle scheme, and a PDMS wafer with a negative surface pattern was processed by Zhong din yu xuan new materials technology corporation (Anhui, China). The PDMS wafer was used to replicate the shape of the microneedle array (10 × 10 microneedles), microneedle base diameter 300 ± 10 μm, microneedle height 800 ± 10 μm, radius of microneedle tip 15 ± 5 μm.

To fabricate a basic SU-8 MN, preparation step was followed: (1) The holder with PDMS wafer of microneedles was under hydrophobic treatment using chlorotrimethylsilane (Aladdin corporation, cas NO. 75-77-4) for 3 min under room temperature. (2) SU-8 2075 photoresist (Micro Chem corporation, USA) was added into the holder, and vacuumed for 3 min, then kept in stationary for 5 min. (3) The whole mould was heated in 90 °C for 90 min. (4) The whole mould was exposed in lithography machine (ABN incorporation, USA) for 480 s. (5) It was heated on a heating plate in 95 °C for 18 min. (6) Holder was removed by cutting. (7) The front side and back side of the remained were exposed for 5 min by lithography machine. (8) The remained was heated on a heat plate in 95 °C for 12 min. (9) A SU-8 microneedle patch was peeled from the mould.

To fabricate conductive three-in-one SU-8 MN, steps are as followed: (i) The basic SU-8 MN patch was insulated by insulating biocompatible silicone glue (Shenzhen Ausland corporation, Shenzhen), and incubated in over in 60 °C for 4 h, and the insulation was protected by 3 M tapes (ii) A compact gold film was formed on its surface by Au spurring (10 mA, 600 s, EMS 150 ion sputtering instrument, USA). (iv) A gold wire (diameter of 200 µm) was attached to the contact area of the SU-8 MN patch by carbon paste (SPI Supplies Co., USA) and placed in the oven (60 °C, 10 min). (v) It was coated a gold film by Au spurring (10 mA, 300 s, EMS 150 ion sputtering instrument, USA) to maintain a consistent surface. (vi) Insulation and package step were applied for the contact and non-conductive area by insulating biocompatible silicone glue (Shenzhen Ausland corporation, Shenzhen), and it was placed in oven (60 °C, 4 h). (vii) For reference and courter compartment, the corresponding MN area was pasted with Ag/AgCl ink (BAS Inc., Japan) and carbon paste (SPI Supplies Co., USA), cured at 80 °C for 30 min.

### Functionalization, characterization of TDN-Ng microneedles

The functionalization steps are as followed: (i) before construction, the conductive MNs was sterilized under UV light for 10 min and plasma-treated for 1 min. (ii) 100 µL 0.2 mg/mL carboxyl graphene (nanoflakes, XFNANO Co., Nanjing) dispersed in 0.1% chitosan was immediately drop-casted on MNs surface and placed in the oven for 30 min under 60 °C. (iii) Carboxyl group of graphene was activated by 200 µL mixed solution of EDC (200 mM): NHS (50 mM) with a volume of 100 µL:100 µL in 100 µL 100 mM 2-morpholinoethanesulfonic acid buffer (all provided by Aladdin Co., Shanghai) for 60 min under 37 °C. The remained solution on the MNs surface was eliminate by compressed air gun. (iv) To stabilized TDN on the surface, the MNs was incubated in 1.6 µM TDN solution for 60 min under 37 °C. The remained solution on the MNs surface was eliminate by compressed air gun. (v) To anchor Ng protein, 15 µL of Ng protein (0.7 µM) was added into the above-mentioned 135 µL EDC/NHS solution, and incubated under 37 °C for 120 min. (vi) 1% BSA blocked the non-specific active sites for 60 min under 37 °C. (vii) The MNs was incubated in 1 µM guide DNA (provided by Sangon corporation, Shanghai) for 60 min under 37 °C. (viii) MNs was washed by 2 mM $MgCl_2$ for 1 min. And it was stored with protection buffer (20% Glycerol in PBS buffer) in -20 °C refrigerator prior to utilization.

For TDN-Ng micro-electrode biosensor, the functionalization step was implemented on a commercial gold electrode (2 mm of diameter) purchased from Gaoss Union incorporation (Wuhan, China), which is the same as the mentioned above.

For mechanical characterization, MN patch was tested on Instron 5966 electronic universal testing machine (Instron, USA) respectively, using 1 kN force sensor (n = 3). The compressing testing mode was under different loading force.

### Preparation and characterization of the TPU patch for iontophoresis

To fabricate basic TPU patch, firstly, a glass petri dish master was hydrophobic treatment by chlorotrimethylsilane (Aladdin corporation, cas NO. 75-77-4) for 3 min under room temperature. The TPU powder (SG-80A, Tecoflex™, Lubrizol corporation, USA) of 1.2 g was mixed with 45 mL N,N-Dimethylformamide (DMF, provided by Aladdin corporation), and the mixture was stirred until TPU powder dispersed. Then, 337 µL tween 20 (Huji corporation, Shanghai) was added into the mixture under sonication for 30 min. The mixture was immediately poured into the glass petri dish master and transferred into oven under 80 °C for 17 h. The TPU film was peeled from the substrate.

To obtain patterned TPU patch, a spray-gun (300 µm diameter, C100, Shibangde corporation, shanghai) spray-deposited different nanomaterials on the surface. Firstly, a metallic microfluidic mask with the desired pattern was gently place on the surface of the TPU film on a heat plate of 60 °C. The silver aerosol ink (Metalon™, NovaCentrix corporation, USA) was spray-printed on the surface through hollow region of the mask and heated for 3 min for evaporating remained ink. Secondly, carbon nanotube (CNT) ink (XFNANO Co., Nanjing) was spray-printed on the TPU surface to form double-layered conduction pattern and heated for 3 min. Finally, a functionalized TPU wearable patch was obtained.

All the mechanical testing was performed on Instron 5966 electronic universal testing machine (Instron, USA). Finite element analysis was conducted by COMSOL Multiphysics 5.3 software. The SEM was conducted on VEGA 3 XMU (TESCAN Co., Czech).

### Ng reaction in vitro assay, molecular diagnosis, and electrophoresis

For Ng reaction in vitro, pre-incubation solution comprised of 7 µL ddH2O, 8 µL working buffer (50 mL total containing 1 mL of 0.02 M Tris-HCl buffer, 1.11 g KCl, 0.005 g $MgCl_2.6H_2O$, 0.005 g $MnCl_2.4H_2O$, 50 µL DDT of 2 M, provided by Sangon corporation, Shanghai), 10 µL guide DNA (10 µM), 15 µL Ng protein (1 µM, provided by Novaprotein corporation), for 60 min, 37 °C. Then, 10 µL target nucleic acid was added for Ng reaction and incubated for 60 min under 37 °C.

For cfDNA standard sample, conserved nucleic acid fragments of the EBV BamHI-W region were screened (GenBank No. A10072.1) and cloned into the PUC57 plasmid, containing the target nucleic acid fragment. And the cfDNA standard sample was obtained through basic PCR assay. Then, the sample was evaluated by next-generation sequencing and nanodrop UV-vis spectrum for measuring its concentration. For RNA standard sample, two conserved nucleic acid fragments of the EBV EBER-2 region (GenBank No. GU205107.1) and EBV LMP 2 A region (GenBank No. M87778.1) were screened and synthesized by GENEWIZ corporation (Soochow, China). For SA and PA standard sample, two conserved nucleic acid fragments of SA strain (GenBank: MZ067393.1) and PA strain (GenBank: D30812.1) were screened, blasted and synthesized by GenScript corporation (Nanjing, China). All the primer (provided by Sangon corporation, Shanghai) and sequences were provided in Supplementary Table 1.

PCR was carried out using a Yeasen Hieff PCR mastermix Kit (Yeasen Biotech Co., Ltd., Shanghai, China). The 50-µL PCR system comprised 25 µL of 2×Hieff Premix plus, 1.5 µL of forward and reverse primers (final concentration of 400 nM), 2 µL of template, and 20 µL of RNase-free ddH2O. The amplification was performed using a fluorescent quantitative PCR detection system (LineGene 9660, Hangzhou Bioer Technology Co., Ltd., Hangzhou, China) according to the following two-step procedure: 1 cycle at 95 °C for 15 min, followed by 40 cycles at 95 °C for 10 s, 60 °C for 20 s, and 72 °C for 32 s.

RT-PCR was carried out using a Yeasen Hifair III one step RT-qPCR SYBR Grenn kit (Yeasen Biotech Co., Ltd., Shanghai, China). The 20-µL RT-PCR system comprised 10 µL of 2×Hifair III SG buffer, 1 µL of Hifair UH Enzyme, 0.4 µL of forward and reverse primers (final concentration of 400 nM), 1 µL of template, and 7.2 µL RNase-free ddH2O. The amplification was performed using a fluorescent quantitative PCR detection system (LineGene 9660, Hangzhou Bioer Technology Co.,

Ltd., Hangzhou, China) according to the following three-step procedure: 1 cycle at 42 °C for 10 min, 1 cycle at 95 °C for 5 min followed by 30-40 cycles at 95 °C for 10 s, 60 °C for 30 s.

For PAGE, the as-prepared gel was purchased from Sangon Corporation (Shanghai). Samples and 6×ficoll gel loading buffer III were pre-mixed for 2 min, then added to the gel. Finally, standard PAGE (1×TAE buffer) was performed with an EPS 300 electrophoresis apparatus (Tanon, Shanghai). Afterwards, PAGE gel was stained in Yeared dye (50 μL stock solution in 50 mL 1×TAE buffer, provided by Yeasen Biotech Co., Ltd., Shanghai, China) for 10-15 min and imaged by 4100 digital gel image processing system (Tanon, Shanghai).

For agarose gel electrophoresis, 2% agarose dyed by Yeared (Yeasen Biotech Co., Ltd., Shanghai, China) was used under an EPS 300 electrophoresis apparatus (1×TAE buffer).

For EMSA, 50 μL target DNA solution containing two single-stranded DNA (100 μM, S1, S2-FAM, provided by Sangon corporation) of 5 μL, 25 μL of ddH₂O, 15 μL of annealing buffer (provided by Beyotime corporation), was denatured at 95 °C for 15 min and annealed to 25 °C by the rate of 1 °C/min (LineGene 9660, Hangzhou Bioer Technology Co., Ltd., Hangzhou, China). Then, the Ng reaction process was the same as mentioned above. And the Ng reaction sample were loaded into a 2% agarose gel without dye for electrophoresis (1×TAE buffer, 100 V). Finally, the gel was exposed under green channel of 4100 digital gel image processing system (Tanon, Shanghai).

## Setup of the wearable electronics

The schematic of flexible circuit board was design by us and fabricated by Refresh AI biosensor corporation (Shenzhen, China). The as-prepared three-in-one microneedles, including working, reference, counter, was attached to the pin that inserted into conductive hole of the flexible circuit board. The prepared TPU patch was connected to the pin on the flexible circuit board. The integrated wearable system was wrapped into a soft package, with the exposure of TPU patch and microneedles. Finally, the real-time monitoring starts to record data by a custom APP on a mobile terminal. The application and usage of the wearables in real-world scenario was shown in supplementary movie 3, 4 and Supplementary Fig. 36.

## Animal experiments

Before animal experiments, skin chip, cell transfection and drug screening were implemented. CNE cells line was purchased from Beina Chuanglian Biology Research Institute (Catalog NO. BNCC341794, Beijing, China). Hela-GFP-Luc cells line was purchased from Quicell corporation (Catalog NO. QuiCell-H548 Shanghai, China). Animals were cared for and maintained under the Guidelines of Laboratory Animals of Fudan University and approved by the Animal Ethics Committee of Fudan University, China (2021JSCHEM-020).

For animal experiment, mice as animal models are chosen and the detailed information is listed as followed: the 4-week-old female Balb/c nude mice (n = 3 for each group) were raised in the housing condition, which is in the cycle of 6-hour dark/18-hour light (23 °C ± 2 °C, 50%-60% of ambient humidity). BALB/c nude female mice (aged 4 weeks) was purchased from Beijing Vital River Laboratory Animal Technology Co., Ltd. (Beijing, China). Female mice were used in this study. The findings didn't apply to only one sex. Sex was not considered in study design. The data disaggregated for sex has not been collected for sex-based analysis, because sex was not relevant to the experiments. For establishing EBV-related model, 500-μL volume of reprogramming CNE-Luc cells ($8 \times 10^8$ cells/mL, 250 μL) and BD matrigel (250 μL, Corning corporation) was subcutaneously injected into 8-week-old BALB/c nude mice. The whole process was conducted under 10 °C. For establishing sepsis-related model, 8-week-old BALB/c nude mice were subjected to tail-vein injection of 60 μL lipopolysaccharide (5 mg/mL, LPS, provided by Sigma-Aldrich), while NTC group was treated with the same volumes of saline. Then, 200 μL of SA and PA strain ($1 \times 10^5$ CFU/mL) was subcutaneously injected into LPS mice.

For bioluminescence, isofluorane anesthesia was maintained using a nose cone delivery system during image acquisition. Mice bearing with CNE-Luc cells and negative control mice were injected with 150 μL of D-Luciferin potassium salt (30 mg/mL, Yeasen Biotech Co., Ltd., Shanghai, China). After 10 min, bioimaging signals was collected by CCD with exposure time of 60 s (in Vivo Xtreme, Bruker, USA).

For TDN-Ng MN patch monitoring, the procedures were as followed: (1) anesthesia was applied to the mice (2 % avertin, 250 μL for each subject). 2 % avertin was prepared as followed: 0.625 g of 2,2,2-tribromoethanol (Sigma-Aldrich) was dissolved in 1.2 mL tert-amyl alcohol (Sinopharm chemical reagent corporation, China), and 30 mL ddH₂O was added. The solution was incubated in water-bath overnight (45 °C) and protected by foil. Mice skin was cleaned by scrub cream and disinfected by 75% ethanol on a heating plate. Then, TE buffer was smeared on the region of interest close to anode side and dried by cotton. The TDN-Ng MN patch was laminated on the region of interest. The work mode for animals consists of 100-second calibration, 3-min reverse iontophoresis, and 100-second detection signal recording. One microneedle patch was used for one mouse during the whole process.

For parallel gold standard PCR method, the ISF of three kinds of mice models were sampled by Whatman card and extraction of nucleic acid was conducted by commercial kit (Sangon provided) before PCR reaction.

## Electrochemical measurements

Differential pulse voltammetry (DPV) testing: scanning potential range was set from -0.2 V to 0.6 V at a scan rate of 50 mV/s in a 0.05 M K₃[Fe(CN)₆]/K₄[Fe(CN)₆] solution that contained 0.50 M KCl. The potential cut off for DPV was at 0.6 V.

Cyclic voltammetry (CV) measurement: scanning potential range was set from -0.2 V to 0.6 V at a scan rate of 50 mV/s in a 0.05 M K₃[Fe(CN)₆]/K₄[Fe(CN)₆] solution that contained 0.50 M KCl.

I-T measurement: sample interval, 0.1 s; quiet time, 2 s; sensitivity (A/V), $1 \times 10^{-4}$ A/V in PBS (0.01 M, pH 7.4). The following equation 1 was used to calculated I response for real-time monitoring signal recording, where $I_t$ and $I_b$ referred to measured data and calibrated background data.

$$I\,response(\%) = \frac{I_t - I_b}{I_b} \times 100\% \qquad (1)$$

## Docking algorithm of the engineered Ng system

Firstly, the structure of Ng protein was constructed via AlphaFold server (deleting the front vector of MGSSHHHHHHSSGLVPRGSH), and optimized by Discovery Studio. Guide DNA was also constructed by Discovery Studio. The model results of Ng protein and guide DNA were visualized by Pymol and Discovery Studio.

Molecular dynamic simulation was based on Gromacs 2020.6 and the detailed was as followed: water molecules model was chosen as SPC/E, and Charmm27 all-atom force field was chosen as the force field. A dodecahedral periodic box was created, and boundary conditions were set in three spatial dimensions. The minimum distance between the protein and the box edge was 1.2 nm. Firstly, the protein-gDNA complex was energy-minimized in vacuum, then the simulation box was filled with water molecules, and sodium and chloride ions were added to mimic the physiological environment, with an ionic concentration of 150 mM, and the number of ions was adjusted to balance the system charge. After adding the solvent, the protein was energy-minimized again, where the backbone atoms of the protein were restrained at fixed positions while the solvent diffused freely. The

coupling temperature is 310 K, and the dispersion was corrected by EnerPres, with Parrinello-Rahman as the pressure controller. The LINCS method was used to constrain all bonds, and the long-range electrostatic interaction method was PME (Particle Mesh Ewald), with an electrostatic cutoff of 1.0 nm. After pre-equilibration simulation, a 100 ns simulation was performed. Subsequently, the root mean square deviation (RMSD) and the number of interaction hydrogen bonds of the protein were calculated by Gromacs 2020.6 program, and further visualized by Xmgrace and Pymol.

## SPR measurement

Before SPR, target cfDNA labeled with biotin was prepared via standard basic PCR assay[67] (Sequence in Supplementary Table 1, provided by Sangon corporation, Shanghai), and immobilized on the surface of Series S Sensor Chip SA (Cat. No. BR-1005-31, Cytiva corporation), using Amine Coupling Kit (Cat. No. BR100050, Cytiva corporation). The analyte of Ng-protein-guide DNA complex was injected into the Chip SA in the association and dissociation flow rate of 30 μL/min (association 60 s, dissociation 150 s) and the regeneration flow rate of 20 μL/min in 30 s. The running buffer was 1×HEPES (10 mM HEPES, 150 mM NaCl, 3 mM EDTA, pH7.4) with 0.005%Tween-20. The affinity constant was calculated by the following Eq. (2), where the affinity constant in the range of $10^{-6}$ to $10^{-9}$ could be judged as the strong binding.

$$K_D = \frac{K_{off}}{K_{on}} \qquad (2)$$

## AFM imaging

For the AFM imaging of different TDNs, the sample preparation was as followed. 40 μL 0.5% APTES (Sigma-Aldrich) was added into the surface of mica for 2 min at room temperature. And the mica was washed by Milli-Q water by pipetting and dried with compressed air-gas. 20 μL of 1 μM different TDNs was added into center of the mica, then 80 μL of TM buffer (TDNs final concentrations of 200 nM) was added, for 15 min at room temperature. Finally, the mica was clean by Milli-Q water by pipetting, and dried with compressed air-gas. All the TDNs sample were operated in ScanAsyst mode in air using super-sharp tips (SNL-10) at a scanning rate of 1.75 Hz.

For the AFM imaging of Ng protein-TDN17 complex on mica, this complex was prepared on tube followed the steps mentioned in the section of Methods 4.5 and the steps of the complex immobilization on the surface of mica were the same as that of TDNs. This complex sample was operated in PeakForce Quantitative Nanomechanical Mapping mode in air using super-sharp tips (SNL-10) at a scanning rate of 1.75 Hz.

## Statistical analysis

All statistical results presented was performed by two-way ANOVA and linear regression analyzed by using origin 2018 software. These results performed were mean ± SD. Significance level was implied by *, **, ***, ****, ns for $p < 0.05$, $p < 0.01$, $p < 0.001$, $p < 0.0001$, no significance respectively.

## Statistics and reproducibility

Data are presented as mean values ± SD and analyzed by two-way ANOVA, calculated using Microsoft Excel 2016 version, where $P$ value significance is also provided for figures. N values are also indicated within figure legends and refer to the independent test on the reproducibility of experiments. The animal sample sizes are chosen by randomization (three animals for each group), without blinding.

## Reporting summary

Further information on research design is available in the Nature Portfolio Reporting Summary linked to this article.

## Data availability

All data supporting the findings of this study are available within the article and its supplementary files. Any additional requests for information can be directed to, and will be fulfilled by, the corresponding authors. Source data are provided with this paper. The GenBank data generated in this study have been deposited in the NCBI database under accession code No. A10072.1, No. GU205107.1, No. M87778.1, MZ067393.1, D30812.1 for EBV BaamHI-W, EBV EBER-2, EBV LMP 2 A, SA strain, PA strain respectively (accessible link, https://www.ncbi.nlm.nih.gov/). Source data are provided with this paper.

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

## Acknowledgements

We gratefully acknowledge the financial support by the National Natural Science Foundation of China (22374029 and 22174024 to X. F.; 22174022 and 22127806 to J. K.), Shanghai Scientific and Technological Innovation Action Plan (23ZR1403200 to X. F.) and National High-level Talent Program (QWH1615018 to X. F.); we also thanks China National Postdoctoral Program for Innovative Talents (BX20200090 to B. Y.), China Postdoctoral Science Foundation Funded Project (2021M700806 to B. Y.) and Shanghai Post-doctoral Excellence Program (2020066 to B. Y.).

## Author contributions

B.Y. and X.F. conceived the idea of the wearable system. B.Y. designed wearable patch, SU-8 microneedles, biosensing strategy, and bio-microfluidics. B.Y. fabricated microneedles and TDN. B.Y. and H.W. conducted electrochemical, bio-microfluidics, electrophoresis, cells, materials characterization, and animal experiments. B.Y. and H.W. applied the microneedles combined with wearable device for animals. B.Y. and X.F. analyzed the data from in vitro and in vivo tests. B.Y. and X.F. wrote the manuscript. X.F. and J.K. supervised the study.

## Competing interests

The authors declare no competing interests.
