## [Peer Review File · Nature Communications]

REVIEWER COMMENTS

Reviewer #1 (Remarks to the Author):

This research has successfully developed a novel wearable device for real-time and continuous monitoring free DNA, particularly leveraging NgAgo to detect specific DNA from the Epstein-Barr virus (EBV) - a known indicator of sepsis. The device's utility in a clinical setting, and its stability over a two-week period, makes it an excellent tool for managing sepsis patients. However, additional research is required to confirm the device's in vivo stability over extended periods and its specificity to target DNA, ensuring it is not influenced by other components or similar DNA sequences. Below are the specific suggestions or concerns:

1. One study observed that the presence of EBV DNA in plasma correlated with the severity of illness, with 32% of septic patients exhibiting detectable levels of EBV DNA in their plasma, compared to only 5% of critically ill, non-septic patients and 0.6% of healthy controls [Walton AH, Muenzer JT, Rasche D, et al. Reactivation of multiple viruses in patients with sepsis. PLoS One. 2014;9(2):e98819]. This raises questions about whether EBV DNA as a good indicator of sepsis. Given that cfDNA or RNA is present in CNE cell lines and it is artificial to use such a model, could the author test mouse sepsis model directly?

Alternatively, other potential applications of this device should also be explored further.

2. Since other proteins or nucleic acid could potentially bind to the Ng or gDNA, the authors should test detection of DNA in the presence of serum or cell lysate in vitro.

3. The specificity of in vivo detection requires testing as well. e.g. sequence similar DNA

4. The response to cfDNA is very sensitive but whether the removal of cfDNA in vitro decreases the signal is unknown. This could also be significant if the device responds to input change.

5. Although the device has been shown to monitor cfDNA for 14 days, there were no observable changes in cfDNA levels. It would be useful to design experiments that test the device's detection capabilities when cfDNA levels decrease or increase over the implantation period.

Reviewer #2 (Remarks to the Author):

This manuscript describes a wearable nucleic acid sensor detecting binding of a target sequence to surface-bound NgAgo. Data are presented to show the devices working in vivo and RNA and DNA species appear to be detectable. Much of the manuscript is focused on the device manufacturing, perhaps

better suited to a field-specific journal, but the overall story is generally interesting. But I feel a few points should be addressed before this can be published.

1) The authors talk about sepsis frequently, but nothing done here has anything to do with sepsis. Detecting EBV is fine but sepsis can be caused by bacteria, fungi, other viruses, and I see no argument that EBV is a universal marker. Other sepsis tests have had to be highly multiplexed, there is not a single "sepsis" marker. Furthermore, the authors talk about sensitivity quite a lot, but a limitation of microvolume testing is that circulating nucleic acids are not present at high concentrations, sometimes even in active infection. In a real-world setting, i.e. not in an artificial mouse model, what concentration of the target nucleic acids would the system actually see with the microneedle sampling? The described sensitivity of 0.3 fM is fine but what is needed/expected in real samples? And the ICU setting is just presented as a given, but is it in fact true that constant monitoring for nucleic acid is useful in this environment? Some more justification to that is needed in my view.

2) The NgAgo characterization is not very convincing, I see no evidence of binding in Figure 3 despite what the authors claim. It would be good to see the data with a scrambled gDNA or something to better demonstrate target-specific performance. And NgAgo is a famously difficult protein to use, but the authors claim it's stable for 3 weeks at room temperature based on the binding data curves, another control or two there with a different guide or something would be helpful. Several times the authors describe potential "scanning" of the NgAgo along target DNA, but to do that would it not have to bind the DNA? I.e. in this sensor setup what would be the difference between "scanning" and target-bound conformations, both would be DNA attached to the NgAgo would they not? I'm curious how this wouldn't just give constant nonspecific signal, and perhaps this could be shown by more data with non-target DNAs.

3) The manuscript could benefit from a thorough edit for readability. The Introduction is largely just lists of not necessarily related previous work. Sentences like "...confronted with limitations, such as long turnover time, labour-intensive, bulky equipment, false-positive, and nonspecific amplification" are barely readable and need some work.

Reviewers' Comments

Reviewer #1 (Remarks to the Author):

This research has successfully developed a novel wearable device for real-time and continuous monitoring free DNA, particularly leveraging NgAgo to detect specific DNA from the Epstein-Barr virus (EBV) - a known indicator of sepsis. The device's utility in a clinical setting, and its stability over a two-week period, makes it an excellent tool for managing sepsis patients. However, additional research is required to confirm the device's in vivo stability over extended periods and its specificity to target DNA, ensuring it is not influenced by other components or similar DNA sequences. Below are the specific suggestions or concerns:

Response: Thank you very much for your review and comments. Hereby we are very delight to answer these concerns point-by-point.

1. One study observed that the presence of EBV DNA in plasma correlated with the severity of illness, with 32% of septic patients exhibiting detectable levels of EBV DNA in their plasma, compared to only 5% of critically ill, non-septic patients and 0.6% of healthy controls [Walton AH, Muenzer JT, Rasche D, et al. Reactivation of multiple viruses in patients with sepsis. PLoS One. 2014;9(2):e98819]. This raises questions about whether EBV DNA as a good indicator of sepsis. Given that cfDNA or RNA is present in CNE cell lines and it is artificial to use such a model, could the author test mouse sepsis model directly? Alternatively, other potential applications of this device should also be explored further.

Response: Thank you for your comments. It's a good question. We are delight to explain this issue and conduct another sepsis mice model experiments and other applications of this device, including

staphylococcus aureus (SA) and pseudomonas aeruginosa (PA).

Firstly, according to some clinical reports, they found that EBV had also been shown to be reactivated in pediatric patients with sepsis (Pediatr. Crit. Care. Med. (2018) 19: e14-22), and some researchers showed that EBV DNA loading was detectable in ICU paediatric patients with sepsis (The Journal of pediatrics, 2019, 213: 82-87. e2). Additionally, Mallet et al. reported that the percentage of patients in sepsis representing EBV reached 33% at the end of the first week in ICU, which the highest among the group, including EBV, HHV, CMV, HSV1. Thus, they suggested that EBV may be useful as additional markers to predict clinical deterioration in ICU patients (Frontiers in Immunology, 2021, 12: 698808). Hence, EBV can be an alternative biomarker for some sepsis patients in ICU, which was illustrated in the main paper as followed.

According to the comments of the reviewer, we tested two other kinds of sepsis mice directly using this device (sepsis mice bearing staphylococcus aureus and pseudomonas aeruginosa, respectively). The experimental streamline was as followed.

We used this TDN-Ng MN patch for longitudinal monitoring other nucleic acid associated with sepsis, including Staphylococcus aureus and Pseudomonas aeruginosa (denoted as SA and PA, respectively). Firstly, for standard samples, two conserved nucleic acid fragments of SA strain (GenBank: MZ067393.1) and PA strain (GenBank: D30812.1) were screened, blasted and synthesized by GenScript corporation (Nanjing, China). All the primer (provided by Sangon corporation, Shanghai) and sequences were provided in Supplementary Table 1. After optimization, SA and PA primers were able to detect a serial of different concentration of target DNA

(Supplementary Figure 37). And the target gene fragments were also detected in SA and PA bacterial strains (Supplementary Figure 37).

Supplementary Figure 37 Dynamic curves and calibration of SA and PA by PCR. (a) and (b) PCR plots for nuc gene fragment of SA. (c) and (d) PCR plots for lasR gene fragment of PA. (e) PCR plots for real SA strain. (f) PCR plots for real PA strain.

We primarily explored other applications of this device in vitro. The standard samples of SA and

PA were chosen for examining practical applications of the device. From the data of Supplementary Figure 41, it showed reliable applications for real-time monitoring other disease-related DNA in vitro.

Supplementary Figure 41. The application of TDN-Ng sensor for real-time monitoring SA and PA target DNA in vitro, PBS (0.01 M, pH 7.4), 37° C, reverse iontophoresis of 10 V.

Then, for constructing sepsis animal models, 8-week-old BALB/c nude mice were subjected to tail-vein injection of 60 μ L lipopolysaccharide (5 mg/mL, LPS, provided by Sigma-Aldrich), while NTC group was treated with the same volumes of saline. Then, 200 μ L of SA and PA strain (1×10^5 CFU/mL) was subcutaneously injected into LPS mice. Based on previous reference (Advanced Healthcare Materials, 2023: 2203133), the inflammatory cytokine storm of sepsis mice was usually within 24h. Thus, we used this patch to monitoring SA and PA sepsis mice within 36h.

As shown in Figure 5g-5h and Supplementary Figure 38-39, this TDN-Ng patch was competent to monitoring two sepsis-related nucleic acid within the period of inflammatory cytokine storm. From the data, the peak values of target gene fragments were 4-hour time point and 12-hour time point for SA and PA sepsis mice, respectively. And the TDN-Ng MN patch had a correlative trend to gold standard PCR. Based on these results, it showcased that the TDN-Ng MN patch was able to

applied for sepsis mice model directly within the period of inflammatory cytokine storm.

Figure 5g Continuous real-time monitoring of SA sepsis mice within 36 hours by TDN-Ng MN patch.

Figure 5h Continuous real-time monitoring of PA sepsis mice within 36 hours by TDN-Ng MN patch.

Figure 38 Raw data of continuous parallel demonstrations on sepsis mice bearing SA and PA strain

at different time points for 36 h.

Supplementary Figure 39. Paralleled demonstrations on sepsis mice bearing SA and PA strain at different time points within 36 h recorded by gold standard PCR.

The detailed revised sentences are listed as followed.

“In this study, we report fully integrated wearable electronics based on tetrahedral DNA nanostructures (TDNs) and prokaryotic argonaute technology for universal nucleic acid real-time monitoring and sepsis-associated intervention caused by EBV, staphylococcus aureus (SA) and pseudomonas aeruginosa (PA).” (section 1, page 4)

“And the TDN-Ng MN platform could also real-time monitor other applications, such as staphylococcus aureus and pseudomonas aeruginosa in Supplementary Figure 41.” (section 2.5,

page 20)

“To further examine the ability of detecting sepsis animal models directly, the TDN-Ng patch with specific gDNA was able to monitoring two kinds of sepsis mice bearing staphylococcus aureus and pseudomonas aeruginosa (denoted as SA and PA, respectively), within the period of inflammatory cytokine storm. Shown in Figure 5g and 5h, the peaks of signal responses were 4-hour time point and 12-hour time point for SA and PA sepsis mice, respectively. And the TDN-Ng MN patch had a correlative trend to gold standard PCR (Supplementary Figure 38 to 39). These results showcased that the TDN-Ng MN patch was able to real-time monitoring longitudinal DNA within the period of inflammatory cytokine storm for sepsis animal models.” (section 2.6, page 24)

“In a statistic^{49,50}, it was reported that EBV-associated nucleic acids might be an alternative biomarker to predict clinical deterioration for sepsis patients in the ICU.” (section 3, page 27)

“For SA and PA standard sample, two conserved nucleic acid fragments of SA strain (GenBank: MZ067393.1) and PA strain (GenBank: D30812.1) were screened, blasted and synthesized by GenScript corporation (Nanjing, China).” (section 4.7, page 33)

“For establishing sepsis-related model, 8-week-old BALB/c nude mice were subjected to tail-vein injection of 60 μ L lipopolysaccharide (5 mg/mL, LPS, provided by Sigma-Aldrich), while NTC group was treated with the same volumes of saline. Then, 200 μ L of SA and PA strain (1×10^5 CFU/mL) was subcutaneously injected into LPS mice.” (section 4.9, page 35)

“For parallel gold standard PCR method, the ISF of three kinds of mice models were sampled by Whatman card and extraction of nucleic acid was conducted by commercial kit (Sangon provided) before PCR reaction.” (section 4.9, page 36)

2. Since other proteins or nucleic acid could potentially bind to the Ng or gDNA, the authors should test detection of DNA in the presence of serum or cell lysate in vitro.

Response: Thank you for your comments. It’s a good question. We agree with you and added experiments to demonstrate it.

The non-specific binding (e.g., protein, non-targeted DNA) might exert influence on sensor signal output. Thus, we investigated the anti-interference by testing detection of target DNA in the presence of cell lysate in vitro. To examine the selectivity of the TDN-Ng platform for different cell lines, the platform was used to detect target nucleic acid in CNE and Hela cell lysate in Figure 40, which showed specificity towards target cell lines with significant difference. The results showed that the sensor had satisfied selectivity in cells lysate in vitro.

The detailed revised sentences are listed as followed.

Figure 40. Selectivity of TDN-Ng platform for different cell lines lysate in vitro. 1×10^7 cell/mL cell, incubated at 37 °C for 30 min, analysed by two-way ANOVA: * $p < 0.05$, ** $p < 0.01$, *** $p < 0.001$, **** $p < 0.0001$, data presented as the mean values \pm SDs, $n=3$ independent experiments, repeated time=3, using 50 mM $[\text{Fe}(\text{CN})_6]^{3-/4-}$, CNE as target group, HeLa as non-target group.

3. The specificity of in vivo detection requires testing as well. e.g. sequence similar DNA

Response: Thank you for your comments. We have added experiments. We tested the specificity in vivo on three different mice models, including PA sepsis mice, SA sepsis mice, and EBV mice. Compared with NTC group (referring to other two non-target models, e.g., if PA as target group, SA and EBV as NTC groups), the target groups showed an obvious signal response with significance difference.

Supplementary Figure 42. Specificity detection in vivo of the TDN-Ng MN patch for three different animal models, at the 4h time point. NTC group referring to other two non-target models.

The detailed revised sentences are listed as followed.

“Additionally, the stable selectivity detection was further investigated in vitro and in vivo by cross-over gDNA sequences and animal models respectively, showing that the target DNA was specifically and evidently bound with NgAgo-gDNA system (Supplementary Figure 40, 42, 43).”

(section 2.5, page 19)

4. The response to cfDNA is very sensitive but whether the removal of cfDNA in vitro decreases the signal is unknown. This could also be significant if the device responds to input change.

Response: Thank you for your comments. As for the current version of the device, its sensing mechanism depends on the specific binding of target DNA. And according to the reference of Joseph Wang’s group (Nat. Biotechnol. 2019, 37, 389-406), it’s known that strong binding affinity was compromise to sensor regeneration (that is for signal input changes, increase or decrease). Hence, it’s hard to direct monitor the signal decrease by this kind of sensor. However, in this study, we used an algorithm to address this problem, where the background signal was calibrated before real-time monitoring every time (denoted as baseline signal). As a result, the sensor could record signal input changes (increase or decrease) within the sensor working lifespan.

5. Although the device has been shown to monitor cfDNA for 14 days, there were no observable changes in cfDNA levels. It would be useful to design experiments that test the device's detection

capabilities when cfDNA levels decrease or increase over the implantation period.

Response: Thank you for your comments. It's a good question. And this comment is similar to the Comment 4. Our response is listed in Comment 4. Additionally, we think that enzyme-based sensor could direct monitor target sample level decrease or increase, such as wearable glucose sensor, which is different from our sensor in terms of sensing mechanism. And we added explanation in the main paper.

The detailed revised sentences are listed as followed.

“The following development of this work will focus on directly recording fluctuating input signal (increase and decrease),” (section 3, page 29)

Reviewer #2(Remarks to the Author):

This manuscript describes a wearable nucleic acid sensor detecting binding of a target sequence to surface-bound NgAgo. Data are presented to show the devices working in vivo and RNA and DNA species appear to be detectable. Much of the manuscript is focused on the device manufacturing, perhaps better suited to a field-specific journal, but the overall story is generally interesting. But I feel a few points should be addressed before this can be published.

Response: Thank you very much for your review and comments. Hereby we are very delight to answer these concerns point-by-point.

(1) The authors talk about sepsis frequently, but nothing done here has anything to do with sepsis. Detecting EBV is fine but sepsis can be caused by bacteria, fungi, other viruses, and I see no argument that EBV is a universal marker. Other sepsis tests have had to be highly multiplexed, there is not a single "sepsis" marker. Furthermore, the authors talk about sensitivity quite a lot, but a limitation of microvolume testing is that circulating nucleic acids are not present at high concentrations, sometimes even in active infection. In a real-world setting, i.e. not in an artificial mouse model, what concentration of the target nucleic acids would the system actually see with the microneedle sampling? The described sensitivity of 0.3 fM is fine but what is needed/expected in real samples? And the ICU setting is just presented as a given, but is it in fact true that constant monitoring for nucleic acid is useful in this environment? Some more justification to that is needed in my view.

Response: Thank you for your comments. We are delight to answer these issues one by one. And we have added other experiments to demonstrate these issues.

Firstly, according to some clinical reports, they found that EBV had also been shown to be reactivated in pediatric patients with sepsis (Pediatr. Crit. Care. Med. (2018) 19: e14-22), and some researchers showed that EBV DNA loading was detectable in ICU paediatric patients with sepsis (The Journal of pediatrics, 2019, 213: 82-87, e2). Additionally, Mallet et al. reported that the percentage of patients in sepsis representing EBV reached 33% at the end of the first week in ICU, which is the highest among the group, including EBV, HHV, CMV, HSV1. Thus, they suggested that EBV may be useful as additional markers to predict clinical deterioration in ICU patients (Frontiers in Immunology, 2021, 12: 698808). Hence, EBV can be an alternative biomarker for some sepsis patients in ICU, which was illustrated in the main paper as followed.

Of course, we have added experiments using two kinds of sepsis mice models caused by bacteria (sepsis mice bearing staphylococcus aureus and pseudomonas aeruginosa, respectively). The detailed procedures were added in the experimental section 4.9, and on the basis of experimental data (Figure 5g-5h, Supplementary Figure 37-39), the TDN-Ng MN platform showed reliability and feasibility for real-time monitoring two kinds of sepsis mice caused by bacteria (staphylococcus aureus and pseudomonas aeruginosa).

Figure 5g Continuous real-time monitoring of SA sepsis mice within 36 hours by TDN-Ng MN patch.

Figure 5h Continuous real-time monitoring of PA sepsis mice within 36 hours by TDN-Ng MN patch.

Figure 38 Raw data of continuous parallel demonstrations on sepsis mice bearing SA and PA strain

at different time points for 36 h.

Figure 39 Paralleled demonstrations on sepsis mice bearing SA and PA strain at different time points within 36 h recorded by gold standard PCR.

Secondly, we would like to explain the concern about the target DNA concentration of sepsis in real-world. On one hand, based on the previous research (Journal of Emergencies, Trauma and Shock, 2008, 1(2): 119), take EBV-related sepsis patients for example, the concentration of target DNA in patients reach 1.1×10^{-15} M. Another research group reported that septic patients had markedly elevated target DNA loads, generally above 1.6×10^{-17} - 1.6×10^{-15} M (PloS one, 2014, 9(6): e98819). And Lin et al. reported that the median concentrations of EBV DNA in patients reached 4.8×10^{-16} M (The New England Journal of Medicine, 2004, 350, 2461-2470). On the other hand, the occurrence of sepsis in ICU is a rapid progress, due to the inflammatory cytokine storm usually within 2-10 days (Advanced Healthcare Materials, 2023: 2203133; Clinical Immunology,

2021, 223: 108652). And the concentration of target DNA was reported to be higher than average level (PloS one, 2014, 9(6): e98819) in this stage of inflammatory cytokine storm. And our TDN-Ng platform is exactly applied for this stage. Thus, the sensitivity of 0.3 fM has been competent to detecting target DNA in real-world sample.

Third, as for the issue of given ICU setting for constant monitoring, we would like to explain it. It was reported that most of sepsis patients would be directly admitted into ICU (Jama, 2014, 312(1): 90-92). In this stage of ICU stay, it was reported that it is essential to increase the frequency of monitoring during the prime time (within 3 days), for constantly monitoring disease progress (Jama, 2016, 315(8): 801-810). And target DNA detection may be useful to predict clinical deterioration in ICU patients, which is a common method (Frontiers in Immunology, 2021, 12: 698808). Therefore, this TDN-Ng platform might offer an alternative for constant real-time monitoring target DNA, which might be a promising prospect. And in this study, we just used sepsis-related animal models for demonstration. Thus, we have revised the sentences about the issue of ICU setting in the main paper as followed.

The detailed revised sentences are listed as followed.

“Hence, wearables might provide an alternative approach for continuous monitoring, e.g., intensive care unit (ICU) patients with sepsis.” (section 1, page 3)

“To further examine the ability of detecting sepsis animal models directly, the TDN-Ng patch with

specific gDNA was able to monitoring two kinds of sepsis mice bearing staphylococcus aureus and pseudomonas aeruginosa (denoted as SA and PA, respectively), within the period of inflammatory cytokine storm. Shown in Figure 5g and 5h, the peaks of signal responses were 4-hour time point and 12-hour time point for SA and PA sepsis mice, respectively. And the TDN-Ng MN patch had a correlative trend to gold standard PCR (Supplementary Figure 38 to 39). These results showcased that the TDN-Ng MN patch was able to real-time monitoring longitudinal DNA within the period of inflammatory cytokine storm for sepsis animal models.” (section 2.6, page 24)

“In a statistic^{49,50}, it was reported that EBV-associated nucleic acids might be an alternative biomarker to predict clinical deterioration for sepsis patients in the ICU.” (section 3, page 27)

“It’s important to increase frequency of monitoring sepsis patients in ICU⁵⁵.” (section 3, page 28)

“In summary, this study proposed wireless, integrated wearable electronics for real-time monitoring of longitudinal nucleic acids for sepsis-related animal models through engineering biosensing interfaces.” (section 3, page 29)

“For establishing sepsis-related model, 8-week-old BALB/c nude mice were subjected to tail-vein injection of 60 μ L lipopolysaccharide (5 mg/mL, LPS, provided by Sigma-Aldrich), while NTC group was treated with the same volumes of saline. Then, 200 μ L of SA and PA strain (1×10^5 CFU/mL) was subcutaneously injected into LPS mice.” (section 4.9, page 35)

“For parallel gold standard PCR method, the ISF of three kinds of mice models were sampled by Whatman card and extraction of nucleic acid was conducted by commercial kit (Sangon provided) before PCR reaction.” (section 4.9, page 36)

(2) The NgAgo characterization is not very convincing, I see no evidence of binding in Figure 3 despite what the authors claim. It would be good to see the data with a scrambled gDNA or something to better demonstrate target-specific performance. And NgAgo is a famously difficult protein to use, but the authors claim it's stable for 3 weeks at room temperature based on the binding data curves, another control or two there with a different guide or something would be helpful. Several times the authors describe potential "scanning" of the NgAgo along target DNA, but to do that would it not have to bind the DNA? I.e. in this sensor setup what would be the difference between "scanning" and target-bound conformations, both would be DNA attached to the NgAgo would they not? I'm curious how this wouldn't just give constant nonspecific signal, and perhaps this could be shown by more data with non-target DNAs.

Response: Thank you for your comments. We are delight to answer these issues one by one.

Firstly, we agree with you and further added experiments about the evidence of binding through three different scrambled guide DNA for target-specific binding. From the cross-over trials, the TDN-Ng sensor showed stable selectivity and evident binding towards specific target DNA. The TDN-Ng platform exhibited evident signal response when binding with its target, while there was weak signal among NTC groups (referring to other two non-target models, e.g., if PA as target group, SA and EBV as NTC groups).

From the data, it was found that the sensor with scrambled gDNA only bound with its target-specific nucleic acid with significant difference, in comparison of non-specific groups. Based on these data (including commercial standard SPR testing of Figure 3c-3e, cell lysate testing of supplementary Figure 40, in vivo animal specific detection of supplementary Figure 42, different scrambled guide DNA testing of supplementary Figure 43), it showed evidence that the NgAgo-gDNA complex selectively bound with target nucleic acid.

Figure 40. Selectivity of TDN-Ng platform for different cell lines lysate in vitro. 1×10^7 cell/mL cell, incubated at 37 °C for 30 min, analysed by two-way ANOVA: * $p < 0.05$, ** $p < 0.01$, *** $p < 0.001$, **** $p < 0.0001$, data presented as the mean values \pm SDs, $n=3$ independent experiments, repeated time=3, using 50 mM $[\text{Fe}(\text{CN})_6]^{3-/4-}$, CNE as target group, HeLa as non-target group.

Supplementary Figure 42. Specificity detection in vivo of the TDN-Ng MN patch for three different animal models, at the 4h time point. NTC group referring to other two non-target models.

Supplementary Figure 43. Selectivity of the sensor with scrambled guide DNA under different nucleic acid of 0.3 nM in vitro. The sensor was incubated in samples for 30 min, analysed by two-way ANOVA: *p<0.05, **p<0.01, ***p<0.001, ****p<0.0001, data presented as the mean values ± SDs, n=3 independent experiments, repeated time=3, using 50 mM [Fe(CN)₆]^{3-/4-}. NTC group referring to other two non-target models.

Secondly, as for the term of “scanning”, thank you very much for your suggestion, and it was

a mistake. We would like to explain and revise it. In this study, the “scanning” means that the guide DNA searches the sequence of the target DNA via Watson-Crick base pairing. Once matched, the target DNA would be bound with NgAgo-guide DNA complex. To be brief, searching sequence of target DNA firstly, bound to NgAgo-guide DNA complex secondly. We have revised it in the main paper as followed.

The detailed revised sentences are listed as followed.

“Under the guidance of gDNA (24-bp 5'-phosphorylated single-stranded DNA), the entire target sequence is searched via Watson-Crick base pairing. Once matched, the immobilized NgAgo-gDNA could bind with the target nucleic acid, which is well-defined stable due to the candidate gDNA.”

(section 2.1, page 5)

“Based on the principle of other pAgo technology, it was speculated that the complex might search the entire nucleic acid sequence under the guidance of gDNA, where a 5'-phosphorylated 24-nt specific sequence matches the target.” (section 2.4, page 13)

“Additionally, the stable selectivity detection was further investigated in vitro and in vivo by cross-over gDNA sequences and animal models respectively, showing that the target DNA was specifically and evidently bound with NgAgo-gDNA system (Supplementary Figure 40, 42, 43).”

(section 2.5, page 19)

(3) The manuscript could benefit from a thorough edit for readability. The Introduction is largely just lists of not necessarily related previous work. Sentences like "...confronted with limitations, such as long turnover time, labour-intensive, bulky equipment, false-positive, and nonspecific amplification" are barely readable and need some work.

Response: Thank you for your comments. We have polished the readability of the paper. We have quoted the related previous work appropriately. After the revision, the streamline of the introduction is alongside this as followed: nucleic acid detection; some established gold standard method (i.e., PCR and genome sequencing); non-amplification methodology; the importance of real-time monitoring of unamplified nucleic acid; our proposed method.

The detailed revised sentences are listed as followed.

"Among these established tools, PCR⁴, isothermal nucleic acid amplification coupled with CRISPR technology⁵, and whole genome sequencing⁶ have been optimized and possessed the ability to provide physiological information indicative of numerous pathologies over the past decades."

(section 1, page 3)

"Despite great advances, PCR and genome sequencing are confronted with limitations, such as long turnover time and bulky equipment." (section 1, page 3)

"In this study, we report fully integrated wearable electronics based on tetrahedral DNA nanostructures (TDNs) and prokaryotic argonaute technology for universal nucleic acid real-time

monitoring and sepsis-associated intervention caused by EBV, staphylococcus aureus (SA) and pseudomonas aeruginosa (PA).” (section 1, page 4)

“The integrated wearable system is employed for real-time tracking of sepsis-related cfDNA and RNA in interstitial fluid (ISF) and therefore expanded possibility in monitoring sepsis for ICU onset.”

(section 1, page 4)

“It’s important to increase frequency of monitoring sepsis patients in ICU⁵⁵.” (section 3, page 28)

REVIEWER COMMENTS

Reviewer #1 (Remarks to the Author):

The authors effectively addressed the majority of the raised questions in their responses. Regarding point 4, an enhancement could be made by suggesting the utilization of DNase to eliminate DNA and subsequently monitor the signal recovery.

Reviewer #2 (Remarks to the Author):

I commend the authors for the updates and additional data included with the revised manuscript, I do think the revised version is better and a more compelling presentation of the sensor.

However I'm still unconvinced by Figure 3a-b which to me do not show any binding of NgAgo to guide or target DNA. If NgAgo is added in excess to 3a why is there barely a shift, and why is it so small considering the size of the protein (supplemental SDS-PAGE has 130kD, but authors say the predicted size is 100, but also 70-80 kD; this should be straightened out). And 3b does not show any shift to me, and would the NgAgo not cut the template strand? Other NgAgo papers show very clear activities on gels, most of the convincing-looking data in this manuscript is derived from interpretation of the sensor binding curves and can show very good looking histograms, but when looked at with standard molecular biology these gels are either not done well or there is no NgAgo binding/activity. If the enzyme is doing what the authors are saying those data should be easy to fix.

Reviewer #1 (Remarks to the Author):

1. The authors effectively addressed the majority of the raised questions in their responses. Regarding point 4, an enhancement could be made by suggesting the utilization of DNase to eliminate DNA and subsequently monitor the signal recovery.

Response: Thank you very much for your review and suggestions again.

We agree with you on this issue. DNase is a good choice for eliminating DNA and the signal might be recovery, which is especially suitable for in vitro detection.

This wearable sensor is expected for continuously real-time on-body monitoring of nucleic acids in vivo, thus it would be more important to balance the DNA elimination and accumulation on the surface of microneedles simultaneously in vivo.

We are now working on another study that can monitor the recovery of signals with implantable biosensing devices (for example wearable indwelling needle). Using the physical shear of blood flow to separate the bound biomolecules, it can real-time record the fluctuation of the signal, such as increase, decrease, recovery (Additional Figure 1). This strategy is still under study, we hope to describe this in detail in future work.

Additional Figure 1. The implantable electronics for the real-time monitoring of target DNA dynamically. (unpublished)

Reviewer #2 (Remarks to the Author):

I commend the authors for the updates and additional data included with the revised manuscript, I do think the revised version is better and a more compelling presentation of the sensor.

However I'm still unconvinced by Figure 3a-b which to me do not show any binding of NgAgo to guide or target DNA. If NgAgo is added in excess to 3a why is there barely a shift, and why is it so small considering the size of the protein (supplemental SDS-PAGE has 130kD, but authors say the predicted size is 100, but also 70-80 kD; this should be straightened out). And 3b does not show any shift to me, and would the NgAgo not cut the template strand? Other NgAgo papers show very clear activities on gels, most of the convincing-looking data in this manuscript is derived from interpretation of the sensor binding curves and can show very good looking histograms, but when looked at with standard molecular biology these gels are either not done well or there is no NgAgo binding/activity. If the enzyme is doing what the authors are saying those data should be easy to fix.

Response: Thank you very much for your comments. We'll explain them one by one.

For the first issue about the size of NgAgo protein, we thank you for your suggestion. We are sorry for our mistake previously. We have confirmed this issue by mass spectrometry which is more convincing. We used mass spectrometry to characterize the size of NgAgo protein in Supplementary Figure 17, and it was 98.2 kDa. And we have revised it in the Supplementary Figure 17 and main paper as followed.

a

b

Protein ID	Master	Accession	Description	Exp. q-val	Contamin	Sum	PEP	S	Coverage	# Peptides	# PSMs	# Unique	# AAs	MW [kDa]	calc. pI	Score	Seq	# Peptides	Protein
High	Master Protein	LOABG6	Stem cell self-renewal protein Piwi domain protein OS=Naionobacterium gregoryi	0	FALSE	105.933	31	24	108	24	887	96.2	4.73	200.58	24	1			
High	Master Protein	A0A238VYG6	DUF1902 domain-containing protein OS=Halorubrum vacuolatum OX=63740 GN=SA	0	FALSE	1.621	8	1	2	1	99	11.5	4.77	0	1	1			
High	Master Protein	A0A238VAA4	Ribokinase OS=Halorubrum vacuolatum OX=63740 GN=SAMN06264855_102127 PE	0	FALSE	1.242	6	1	1	1	382	39.2	4.34	0	1	1			
High	Master Protein	D312D6	Virus protein phiCh1-VP84 OS=Natrialba magadii (strain ATCC 43099 / DSM 3394 / C	0	FALSE	1.208	3	1	1	1	254	29.8	4.55	0	1	1			
High	Master Protein	ADA1H1AQJ8	Probable RNA pseudouridine synthase D OS=Naionobacterium texcoconense OX=	0	FALSE	1.097	9	1	1	1	450	49.9	4.6	0	1	1			
High	Master Protein	A0A238W968	Alpha-galactosidase OS=Halorubrum vacuolatum OX=63740 GN=SAMN06264855_1	0	FALSE	1.094	2	1	1	1	667	72.8	4.59	0	1	1			
High	Master Protein	D35U11	DEAD/DEAH box helicase OS=Natrialba magadii (strain ATCC 43099 / DSM 3394 / C	0	FALSE	1.009	2	1	2	1	785	86.1	4.67	1.75	1	1			
High	Master Protein	A0A1I3J2Q4	PAS domain 5-box-containing protein OS=Naionobacterium gregoryi OX=44930 GN	0	FALSE	0.997	2	1	2	1	637	70.6	4.45	0	1	1			
High	Master Protein	LOALW1	Lycopene cyclase domain protein OS=Naionobacterium gregoryi (strain ATCC 4309	0	FALSE	0.982	3	1	6	1	263	29.3	6.93	0	1	1			
High	Master Protein	A0A1I3J3N8	Cytochrome c oxidase assembly protein subunit 15 OS=Naionobacterium gregoryi	0.009	FALSE	0.957	2	1	1	1	284	30.3	9.5	0	1	1			
High	Master Protein	ADA1H1A8F9	Polyketide cyclase / dehydrase and lipid transport OS=Naionobacterium texcocon	0.009	FALSE	0.947	4	1	1	1	181	20.8	4.88	0	1	1			
Medium	Master Protein	A0A238VLP2	Glycerol-1-phosphate dehydrogenase [NAD(P)H] OS=Halorubrum vacuolatum OX=6	0.018	FALSE	0.92	14	1	1	1	348	36.6	4.83	0	1	1			
Medium	Master Protein	ADA1H1I1Q1	BFN domain-containing protein OS=Naionobacterium texcoconense OX=1095778 C	0.018	FALSE	0.871	6	1	1	1	155	16.8	4.25	0	1	1			
Medium	Master Protein	LY9176	Transposase (Fragment) OS=Naionobacterium gregoryi (strain ATCC 43099 / DSM	0.018	FALSE	0.856	5	1	2	1	134	14.7	6.79	0	1	1			
Medium	Master Protein	ADA1H0YWK6	Creatinine amidohydrolase OS=Naionobacterium texcoconense OX=1095778 GN=5	0.026	FALSE	0.823	13	1	1	1	245	26.3	4.63	0	1	1			

Supplementary Figure 17. Characterization of the engineered NgAgo protein. (a) Secondary mass spectrometry of partial unique peptide. (b) Analysis results predicted from the mass spectrometry.

For the second issue, as for these two figures (figure 3a-3b in previous version), we would like to added other experiments to further demonstrate it. In fact, the previous two figures were PAGE gel (figure 3a in previous version) that dyed by YearedTM dye and agarose gel that dyed by FAM probe (figure 3b in previous version) respectively, which was only for staining and displaying nucleic acid. Thus, what we can see is only the bands of nucleic acid, and protein band can not be seen in these gels. According to the comments of the reviewer, we would like to conduct further tests to show the shift bands, to more clearly demonstrate the binding of NgAgo to guide and target DNA.

Firstly, we would like to explain the mechanism of this method used in our job. Electrophoretic mobility shift assay (EMSA) is one of the methods for studying the DNA-binding properties of a protein, which can be used to demonstrate the binding and relative affinities of a protein. The protein-DNA complexes are separated from free (unbound) DNA by electrophoresis. The protein retards the mobility of the DNA fragments to which it binds; thus, the free DNA migrates faster through the gel than does the DNA-protein complex. The gel reveals the different positions of the free and bound DNA (Nat. Methods, 2005, 2, 557-558). We can see that there were band shift and band drag shadow when the DNA bound with protein in Additional Figure 2 (BioTechniques, 2019, 68, 101-105; Nature protocols, 2007, 2, 1849-1861).

Additional Figure 2. EMSA of other previous reports. (a) Excerpted from reports of BioTechniques, 2019, 68, 101-105. (b) Excerpted from reports of Nature protocols, 2007, 2, 1849-1861.

Then, hereby, we conducted EMSA experiments in our job to demonstrate the binding of NgAgo to guide and target DNA. The detail results were shown in Supplementary Figure 44 and detailed processes were listed in Experimental section 4.7. Guide DNA and target DNA were labeled with FAM probe. From the EMSA gel, we could see a band shift (bound DNA), indicating that the guide DNA had successfully bound to the NgAgo protein (Supplementary Figure 44a). Next, we conducted the testing on the binding between NgAgo/guide DNA complex and target DNA. From the Supplementary Figure 44b, a band shift of NgAgo/guide DNA system appeared (in blue short-dash box), which indicating the binding event.

In this job, we further take dCas9/sgRNA system as an example and comparison, which is recognized as a system that can bind to target DNA without question. From the result (Supplementary Figure 44c), a band shift of dCas9/sgRNA system appeared (in blue short-dash box). Due to its larger size of dCas9 (~160 KDa) than NgAgo (~98.2 KDa). The moving velocity of shift band in CRISPR-dCas9 group is much slower.

Supplementary Figure 44. Electrophoretic mobility shift assay (EMSA) of Ng system. (a) EMSA of guide DNA with NgAgo protein, 2% agarose of no dye, 100V, 35min, $1 \times$ TAE buffer, exposure time of 6s100ms. (b) EMSA of Ng system, 2% agarose of no dye, 100V, 45min, $1 \times$ TAE buffer, exposure time of 6s100ms. (c) EMSA of CRISPR-dCas9, 2% agarose of no dye, 100V, 45min, $1 \times$ TAE buffer, exposure time of 6s100ms.

Above all, this results from the standard molecular biology demonstrated clearly the binding event between the NgAgo and gDNA or target DNA. However, this EMSA approach is only for preliminary qualitative studies of NgAgo/guide DNA binding events to target DNA. We successively used other methods in our job to investigate binding events quantitatively, including SPR (binding constant K_D of 5.49×10^{-9}) and electrochemical analysis in Additional Figure 3. All these results indicated the effective binding of NgAgo protein to guide DNA or target DNA.

Additional Figure 3. Other techniques to demonstrate the binding of NgAgo/gDNA and target DNA. (a) Surface plasmon resonance results of Ng system. (b) Electrochemical measurements of Ng

system.

The detailed experimental processes and revised sentences were listed as followed.

“In Supplementary Figure 44, from the electrophoretic mobility shift assay (EMSA), the bands of binding events had an obvious drag shift. Additionally, through the EMSA, the band of the Ng reaction had a drag shift, which is analogous to CRISPR-dCas9, because the FAM-labelled target DNA had a slower mobility with the binding of proteins. The above results primarily confirmed the recognition between the NgAgo-gDNA complex and target DNA.” (main paper, section 2.4)

“NgAgo is predictably smaller size than that of CRISRP effector (e.g., Cas9, Cas12a)” (main paper, section 2.4)

“For Ng reaction in vitro, pre-incubation solution comprised of 7 μ L ddH₂O, 8 μ L working buffer (50 mL total containing 1 mL of 0.02 M Tris-HCl buffer, 1.11g KCl, 0.005g MgCl₂·6H₂O, 0.005g MnCl₂·4H₂O, 50 μ L DDT of 2 M), 10 μ L guide DNA (10 μ M), 15 μ L Ng protein (1 μ M, provided by Novaprotein corporation), for 60 min, 37°C. Then, 10 μ L target nucleic acid was added for Ng reaction and incubated for 60 min under 37°C.” (main paper, section 4.7)

“For EMSA, 50 μ L target DNA solution containing two single-stranded DNA (100 μ M, S1, S2-FAM, provided by Sangon corporation) of 5 μ L, 25 μ L of ddH₂O, 15 μ L of annealing buffer (provided by Beyotime corporation), was denatured at 95°C for 15 min and annealed to 25°C by the rate of 1°C/min (LineGene 9660, Hangzhou Bioer Technology Co., Ltd., Hangzhou, China). Then, the Ng reaction process was the same as mentioned above. And the Ng reaction sample were loaded into a 2% agarose gel without dye for electrophoresis (1×TAE buffer, 100V). Finally, the gel was exposed under green channel of 4100 digital gel image processing system (Tanon, Shanghai).” (main paper, section 4.7)

REVIEWERS' COMMENTS

Reviewer #2 (Remarks to the Author):

I took a look at the response and additional data, and I'm happy with that update, they addressed my key complaints and the new data are more convincing.